



# Development and evaluation of an ozone deposition scheme for coupling to a terrestrial biosphere model

Martina Franz[1,2], David Simpson[4,5], Almut Arneth[6], and Sönke Zaehle[1,3]

[1]Biogeochemical Integration Department, Max Planck Institute for Biogeochemistry, Jena, Germany
[2]International Max Planck Research School (IMPRS) for Global Biogeochemical Cycles, Jena, Germany
[3]Michael Stifel Center Jena for Data-driven and Simulation Science, Jena, Germany
[4]EMEP MSC-W, Norwegian Meteorological Institute, Oslo, Norway
[5]Dept. Earth & Space Sciences, Chalmers University of Technology, Gothenburg, Sweden
[6]Karlsruhe Institute of Technology (KIT), Department of Atmospheric Environmental Research (IMK-IFU),
Garmisch-Partenkirchen, Germany

*Correspondence to:* Martina Franz (mfranz@bgc-jena.mpg.de)

**Abstract.**

Ozone is a toxic air pollutant that can damage plant leaves and substantially affect the plant's gross primary production (GPP) and health. Realistic estimates of the effects of tropospheric anthropogenic ozone on GPP are thus potentially important to assess the strength of the terrestrial biosphere as a carbon sink. To better understand the impact of ozone damage on the

terrestrial carbon cycle, we developed a module to estimate ozone uptake and damage of plants for the state of the art global terrestrial biosphere model OCN. Our approach accounts for ozone damage by calculating (a) ozone transport from the free troposphere to leaf level, (b) ozone flux into the leaf, and (c) ozone damage of photosynthesis as a function of the accumulated ozone uptake over the life-time of a leaf.

A comparison of modelled canopy conductance, GPP, and latent heat to FLUXNET data across European forest and grass-

land sites shows a general good performance of OCN. In comparison to literature values, we demonstrate that the new model version produces realistic stomatal flux ratios as well as ozone surface resistances and deposition velocities. A sensitivity study reveals that key metrics of the air-to-leaf ozone transport and ozone deposition, in particular the stomatal ozone update are reasonably robust against uncertainty in the underlying parameterisation of the deposition scheme. Correctly estimating canopy conductance plays a pivotal role in the estimate of cumulative ozone uptake.

When applied at the European scale, we find that the added complexity of the ozone uptake simulation substantially affects simulated ozone uptake and accumulation, because aerodynamic resistance and non-stomatal ozone destruction reduce the predicted ozone concentrations outside the leaves. Ozone impacts on GPP and transpiration in a Europe-wide simulation indicate that tropospheric ozone impacts the regional carbon and water cycling less than expected from previous studies.

## 1 Introduction

Tropospheric ozone ($O_3$) is a highly reactive and toxic gas. It enters the plants mainly through the stomata of the leaf, where it forms reactive oxygen species (ROS) which have the potential to damage the leaf. While leaves possess physiological pathways



to produce compounds like ascorbate and polyamines, which help to neutralise the oxidising power of ROS (Kronfuß et al., 1998; Kangasjärvi et al., 1994; Tausz et al., 2007), ozone injury may occur when the leaf's anti-oxidant system becomes overwhelmed (Wieser and Matyssek, 2007).

In Western Europe, tropospheric ozone levels have increased approximately by a factor of 5 from pre-industrial values to the 1990s (Cooper et al., 2014; Marenco et al., 1994; Staehelin et al., 1994) (although the low values at the start of this period are very uncertain) and approximately doubled between 1950 and 1990s in the northern hemisphere (Parrish et al., 2012; Cooper et al., 2014). The major causes for this increased $O_3$ formation is the increased emission of $O_3$ precursor trace gases such as $NO_x$ and CO, primarily from combustion sources, and methane emissions from agriculture and industry (Fusco and Logan, 2003; Vingarzan, 2004). For instance, in Western Europe, $NO_x$ emissions have risen by a factor of 4.5 between 1955 and 1985 (Staehelin et al., 1994). In addition, downward transport of $O_3$ from the stratosphere to the troposphere (Vingarzan, 2004; Young et al., 2013) and intercontinental transport from polluted to less polluted areas (Vingarzan, 2004; Jenkin, 2008; Fiore et al., 2009) can increase local and regional $O_3$ concentrations.

A commonly observed consequence of elevated levels of ozone exposure is a decline in net photosynthesis (Morgan et al., 2003; Wittig et al., 2007), which may result from the damage of the photosynthetic apparatus or increased respiration due to the production of defence compounds and investments in injury repair (Wieser and Matyssek, 2007; Ainsworth et al., 2012). The reduction in net photosynthesis results in reduced growth and hence a reduced leaf area, and plant biomass (Morgan et al., 2003; Lombardozzi et al., 2013; Wittig et al., 2009). The tight coupling between photosynthesis and stomatal conductance further affects canopy conductance, and thereby transpiration rates (Morgan et al., 2003; Wittig et al., 2009; Lombardozzi et al., 2013), likely affecting the ecosystem water balance.

Due to its phytotoxic effect, elevated $O_3$ levels as a consequence of anthropogenic air pollution may affect the land carbon cycle, and potentially reduce the net land carbon uptake capacity (Sitch et al., 2007; Arneth et al., 2010; Simpson et al., 2014a), which currently corresponds to about a quarter of the anthropogenic fossil fuel emissions as a consequence of a sustained imbalance between photosynthetic carbon uptake and carbon loss through respiration and disturbance processes (Le Quéré et al., 2015). However, the extend to which $O_3$ affects plant health regionally and thereby alters terrestrial biogeochemistry and the terrestrial water balance is still subject of large uncertainty (Simpson et al., 2014a).

A number of $O_3$ exposure indices have been proposed to assess the potential detrimental effect of tropospheric $O_3$ on the plants (LRTAP-Convention, 2010; Mills et al., 2011b). A widely used example of these indices is the concentration-based AOTX [ppb h] (accumulated $O_3$ concentration over a threshold of $X$ ppb), which relates the free-air $O_3$ concentration to observed plant damage. Models assessing $O_3$ damage to gross or net primary production based on AOTX have been used for many years and indicate that substantial reduction in plant growth and carbon sequestration occurs globally and may reach reductions of more than 40% at $O_3$ hot spots (Felzer et al., 2004, 2005; Ren et al., 2011; Anav et al., 2011).

A significant caveat of concentration-based assessments of ozone toxicity effects is that species and their regional provenances differ vastly in their canopy conductance. Stomatal control of the leaf gas exchange regulates photosynthesis, and varies inter alia with plant specific photosynthetic capacity and intrinsic water-use efficiency of photosynthesis, phenology, as well as environmental factors such as incident light, atmospheric vapour pressure deficit (VPD), air temperature. The consequent



differences in stomatal conductance implies that the actual ozone dose, and thus the level of ozone-related damage, differs between species exposed to similar atmospheric $O_3$ concentrations (Wieser and Havranek, 1995). The ozone dose, that is the integral of the instantaneous $O_3$ stomatal flux over a given period of time, has been observed to strongly correlate with the amount of injury of a plant, suggesting that plants with higher stomatal conductance are subject to higher doses and hence

more susceptible to injury (Reich, 1987; Wittig et al., 2009).

Accounting for the ozone dose rather than the $O_3$ exposure in assessments of $O_3$ damage results in diverging regional patterns of $O_3$ damage, as regions with the highest exposure ($O_3$ concentrations) do not always coincide with regions of high uptake (Emberson et al., 2000; Mills et al., 2011a; Simpson et al., 2007a). Regions with low AOT40 values might show moderate to high values of $O_3$ uptake because the flux approach accounts for climatic conditions that enable high stomatal

conductances and hence high values of $O_3$ uptake (Mills et al., 2011a).

When calculating the $O_3$ uptake into the plants, it is important to consider that stomatal uptake is not the only surface sink of ozone. Ozone destruction also occurs at non-stomatal surfaces such as the leafs cuticle and soil surface. The stomatal flux represents approximately half of the total $O_3$ flux to the surface (Gerosa et al., 2004; Fowler et al., 2009; Cieslik, 2004; Simpson et al., 2003). Accounting for this non-stomatal $O_3$ deposition reduces the amount of $O_3$ uptake into the plants by reducing the

surface $O_3$ concentration (Tuovinen et al., 2009b) and thus has the potential to affect flux-based ozone damage estimates.

A further challenge in estimating plant damage related to ozone uptake is that plants differ in their ability to remove any ROS from the leaf before damage of leaf cellular organs is incurred (Luwe and Heber, 1995). Conceptionally, one can describe the capacity as a plant-specific $O_3$ dose, with which the anti-oxidant system of the leaves can cope such that no damage is observed (Musselman et al., 2006). Ozone damage is only incurred, once the $O_3$ flux into the leaf exceeds this dose. A

commonly used index to assess flux-based damage to plants is the PODY [Phytotoxic Ozone Dose, $nmol\,m^{-2}\,s^{-1}$], which gives the accumulated ozone flux above a threshold of Y $nmol\,m^{-2}\,s^{-1}$ for all daylight hours and a given time period. Common threshold values for PODY range from 1-6 $nmol\,m^{-2}\,s^{-1}$ (Pleijel et al., 2007; LRTAP-Convention, 2010; Mills et al., 2011b).

Only a few terrestrial biosphere models have adopted the flux approach to relate ozone exposure to plant damage and thus estimate ozone induced reductions in terrestrial carbon sequestration in a process-based manner. Sitch et al. (2007) developed

a version of the JULES model in which stomatal ozone uptake directly affects net primary production (NPP), thereby ignoring the effect of reduced photosynthesis under elevated levels of $O_3$ on water fluxes. Lombardozzi et al. (2015) proposed a revised version of the CLM model, in which $O_3$ imposes fixed reductions to net photosynthesis independent of the amount of cumulated $O_3$ uptake for two out of three modelled plant types.

In this paper, we present a new, globally applicable model to calculate $O_3$ uptake and damage in a process-oriented manner,

coupled to the terrestrial energy, water, carbon and nitrogen budget of the OCN terrestrial biosphere model (Zaehle and Friend, 2010).

In this model, the canopy $O_3$ abundance is calculated using aerodynamic resistance and surface resistances to soil surface, vegetation surfaces and stomatal cavities to take account of non-stomatal $O_3$ destruction. Canopy $O_3$ abundance is used to simulate stomatal $O_3$uptake given instantaneous values of net photosynthesis and stomatal conductance. Ozone uptake and its



effect on net photosynthesis is then calculated based on an extensive meta-analysis across 28 tree species by Wittig et al. (2007) considering the ability of plants to detoxify a proportion of the $O_3$ dose (Sitch et al., 2007).

We first give a detailed overview of the ozone scheme (Section 2.1); evaluate modelled gross primary production (GPP), canopy conductance, latent heat fluxes and LAI against data from the FLUXNET database (Baldocchi et al., 2001) to test the

ability of the model to simulate observed values of key components affecting calculate $O_3$ uptake (Section 3.1); evaluate the simulated ozone metrics against reported values in the literature (Section 3.2); provide a sensitivity analysis of the model to evaluate the reliability of simulated values of $O_3$ uptake (Section 3.3); give an estimate of the effect of the present-day $O_3$ burden on European GPP and transpiration(Section 3.4); and estimate the impact of using the $O_3$ deposition scheme on uptake and accumulation (Section 3.5).

## 2   Methods

We developed an ozone deposition and leaf-uptake module for the terrestrial biosphere model OCN (Zaehle and Friend, 2010). OCN is an extension of the land-surface-scheme ORCHIDEE (Krinner et al., 2005), which simulates the terrestrial coupled carbon, nitrogen (N) and water cycles for twelve plant functional types driven by climate data, atmospheric composition (N deposition, as well as atmospheric $CO_2$ and $O_3$ burden), and land use information (land cover and fertiliser application).

In OCN net photosynthesis is calculated for shaded and sun-lit leaves in a multi-layer canopy following a modified Farquhar-scheme and considering the light profiles of diffuse and direct radiation. Photosynthetic capacity depends on leaf nitrogen concentration and leaf area, which are both affected by ecosystem available N. Increases in leaf nitrogen content enable higher net photosynthesis and higher stomatal conductance per unit leaf area. This in turn affects transpiration as well as ozone uptake and ozone damage estimates.

The ozone and N-deposition data used for this study are provided by the EMEP MSC-W chemical transport model (CTM) (Simpson et al., 2012a). The ozone flux and deposition modules used in the EMEP model are rather advanced compared to most CTMs, and have documented in a number of papers (Emberson et al., 2001; Tuovinen et al., 2004a, 2009a; Simpson et al., 2007b, 2012b; Klingberg et al., 2008). The ozone deposition scheme for OCN is adapted from the model used by the Meteorological Synthesizing Centre - West of the European Monitoring and Evaluation Programme (EMEP MSC-W) (Simpson

et al., 2012a) to fit the land surface characteristics and process descriptions of the ORCHIDEE model.

### 2.1   Ozone module

The ozone deposition scheme calculates ozone deposition to the leaf surface from the free atmosphere, represented by the $O_3$ concentration at the lowest level of the atmospheric chemistry transport model (CTM), taken to be at 45 m above the surface. The total $O_3$ dry deposition flux ($F_g$) to the ground surface is calculated as

$$F_g = V_g \chi_{atm}^{O_3} \tag{1}$$




where $\chi_{atm}^{O_3}$ is the $O_3$ concentration in 45 m height and $V_g$ is the deposition velocity at that height. In OCN $V_g$ is taken to be dependent on the aerodynamic resistance ($R_a$), canopy-scale quasi-laminar layer resistance ($R_b$) and the compound surface resistance ($R_c$) to ozone deposition.

$$V_g = \frac{1}{R_a + R_b + R_c}. \tag{2}$$

5   $R_b$ is calculated from the friction velocity ($u_*$) as

$$R_b = \frac{6}{u_*}. \tag{3}$$

The $R_a$ in 45 m height is not computed by OCN and is inferred from the logarithmic wind profile (for more details see Appendix A). $R_c$ is calculated as the sum of the parallel resistances to stomatal/canopy ($1/G_c^{O_3}$) and non-stomatal ozone uptake ($1/G_{ns}$) (Simpson et al., 2012a, eq. 55)

$$R_c = \frac{1}{G_c^{O_3} + G_{ns}}. \tag{4}$$

The stomatal conductance to ozone $G_{st}^{O_3}$ ($\mathrm{m\,s^{-1}}$) is computed by OCN (Zaehle and Friend, 2010) as:

$$G_{st}^{O_3} = g_1 \frac{f(\Theta)f(q_{air})f(C_i)f(height)A_{n,sat}}{1.51} \tag{5}$$

where $G_{st}^{O_3}$ is calculated as a function of net photosynthesis at saturating $C_i$ ($A_{n,sat}$) where $g_1$ is the intrinsic slope between $A_n$ and $G_{st}$. It further depends on a number of scalars to account for the effect of soil moisture ($f(\Theta)$), water transport limitation with canopy height ($f(height)$), and atmospheric drought ($f_{(q_{air})}$), as well as an empirical non-linear sensitivity to the leafs internal $CO_2$ concentration ($f(C_i)$), all as described in (Friend and Kiang, 2005). The factor 1.51 accounts for the different diffusivity of ozone to water vapour (Massman, 1998). The canopy conductance to ozone $G_c^{O_3}$ is calculated by summing the $G_{st}^{O_3}$ of all canopy layers. To yield reasonable conductance values in OCN compared to FLUXNET data (see Sect. 3.1), the original intrinsic slope between $A_n$ and $G_c$ called $\alpha$ in Friend and Kiang (2005) is adapted such that $g_1 = 0.7\alpha$.



The non-stomatal conductance $G_{ns}$ follows the EMEP approach (Simpson et al., 2012a, eq. 60) and represents the $O_3$ fluxes between canopy air space and surfaces other than the stomatal cavities. The model accounts for ozone destruction on the leaf surface ($r_{ext}$), within-canopy resistance to ozone transport ($R_{inc}$), and ground surface resistance ($R_{gs}$)

$$G_{ns} = \frac{SAI}{r_{ext}} + \frac{1}{R_{inc} + R_{gs}} \tag{6}$$

where the surface area index $SAI$ is equal to the leaf area index $LAI$ for herbaceous PFTs (grasses and crops), and $SAI = LAI + 1$ for tree PFTs according to Simpson et al. (2012a), to account for ozone destruction on branches and stem. Unlike EMEP, we do not apply a day of the growing season constraint for crop exposure to $O_3$, which in OCN is accounted for by the simulated phenology and seasonality of photosynthesis. The external leaf-resistance ($r_{ext}$) per unit surface area is calculated as

$$r_{ext} = r_{ext,b} F_T \tag{7}$$

where the base external leaf-resistance ($r_{ext,b}$) of $2500 \, \mathrm{m \, s^{-1}}$ is altered by $F_T$ a correction factor for low temperatures and

$$F_T = e^{-0.2(1+T_s)} \tag{8}$$

with $1 \leq F_T \leq 2$ and $T_s$ the 2 m air-temperature ($^\circ C$ Simpson et al., 2012a, eq. 60). The within-canopy resistance ($R_{inc}$) is calculated as

$$R_{inc} = bSAI\frac{h}{u_*} \tag{9}$$

where $b$ is an empirical constant (set to $14 \, \mathrm{s^{-1}}$) and h is the canopy height in $\mathrm{m}$. The ground-surface resistance $R_{gs}$ is calculated as

$$R_{gs} = \frac{1 - 2f_{snow}}{F_T \hat{R}_{gs}} + \frac{2f_{snow}}{R_{snow}} \tag{10}$$

(Simpson et al., 2012a, eq. 59). $\hat{R}_{gs}$ represents base-values of $R_{gs}$ and takes values of $2000 \, \mathrm{s \, m^{-1}}$ for bare soil, $200 \, \mathrm{s \, m^{-1}}$
for forests and crops and $1000 \, \mathrm{s \, m^{-1}}$ for non-crop grasses (Simpson et al., 2012a, Suppl.). Like in EMEP, the ground-surface





resistance of ozone to snow ($R_{snow}$) is set to a value of $2000\,\mathrm{s\,m^{-1}}$ according to (Zhang et al., 2003). $f_{snow}$ is calculated from the actual snow depth ($s_d$) simulated by OCN, and the maximum possible snow depth ($s_{d,max}$):

$$f_{snow} = \frac{s_d}{s_{d,max}} \qquad (11)$$

with the constraint of $0 \leq f_{snow} \leq 0.5$. The upper border prevents negative values in the first fraction of eq. 10. $s_{d,max}$ is taken to be $10\,\mathrm{kg\,m^{-2}}$ (Ducoudré et al., 1993).

Given these resistances, the canopy $O_3$ concentration ($\chi_c^{O3}$, $\mathrm{nmol\,m^{-3}}$) is then calculated based on a constant flux assumption

$$\chi_c^{O3} = \chi_{atm}^{O3}\left(1 - \frac{R_a}{R_a + R_b + R_c}\right). \qquad (12)$$

$\chi_c^{O3}$ and the stomatal conductance to ozone ($G_{st}^{O_3}$ in $\mathrm{m\,s^{-1}}$) are used to calculate the ozone flux into the leaf cavities ($F_{st}$, $\mathrm{nmol\,m^{-2}\,s^{-1}}$):

$$F_{st} = (\chi_c^{O3} - \chi_i^{O3})G_{st}^{O_3}. \qquad (13)$$

According to Laisk et al. (1989) the leaf internal $O_3$ concentration ($\chi_i^{O3}$) is assumed to be zero.

It should be noted that the OCN implementation of deposition and flux described above is a simplification of the deposition system used by EMEP. The external leaf resistance is not included in the calculation of $F_{st}$ (Tuovinen et al., 2007, 2009b) what results in an overestimation of stomatal $O_3$ uptake. Further, OCN's calculation of $R_a$ is based upon neutral stability conditions (see Appendix), whereas the EMEP model makes use of rather detailed stability correction factors. However, a series of calculations with the full EMEP model have shown that the uncertainties associated with these simplifications are small, typically less than $1\,\mathrm{mmol\,m^{-2}}$.

## 2.2 Relating stomatal uptake to leaf damage

To estimate the ozone related damage due to stomatal $O_3$ uptake, a flux threshold ($F_{detox}$) is used to account for the plants ability to detoxify part of the ozone.

$$F_{st,detox} = \mathrm{MAX}(F_{st} - F_{detox}, 0) \qquad (14)$$





where the detoxification threshold $F_{detox}$ is set to $1.6\,\mathrm{nmol\,m^{-2}\,s^{-1}}$ for forests and to $5\,\mathrm{nmol\,m^{-2}\,s^{-1}}$ for grasses and crops (Sitch et al., 2007). The function MAX prevents negative uptake values when $F_{st} < F_{detox}$. An accumulation of $F_{st,detox}$ over time gives the accumulated uptake of ozone for a particular canopy layer ($CUO_l$, $\mathrm{mmol\,m^{-2}}$), or for $l = 1$ (top canopy layer) the phytotoxic ozone dose, ($POD_l$, $\mathrm{mmol\,m^{-2}}$)

$$CUO_l = CUO_l(1 - f_{shed}) + cF_{st,detox}\Delta t \tag{15}$$

where $\Delta t = 1800$ seconds is the length of simulation time step and $c = 10^{-6}$ converts from $\mathrm{nmol}$ to $\mathrm{mmol}$.

The phenology of leaves is accounted for by assuming that emerging leaves are undamaged, and by reducing the $CUO_l$ by the fraction of new developed leaves per time step and layer ($f_{shed}$). Furthermore deciduous PFTs shed all CUO at the end of the growing season and grow undamaged leaves the next spring. Evergreen PFTs shed proportionate amounts of CUO during the entire year always when new leaves are grown.

The $CUO_l$ is used to approximate the damage to net photosynthesis ($A_n$) by using the damage relationship of Wittig et al. (2007):

$$d_l^{O3} = \frac{0.22CUO_l + 6.16}{100} \tag{16}$$

where the factor 100 scales the percentage values of damage to fractions. Net photosynthesis accounting for ozone damage ($A_n^{O3}$) is then calculated by subtracting the damage fraction from the undamaged value of $A_n$:

$$A_{n,l}^{O3} = A_{n,l}(1 - d_l^{O3}). \tag{17}$$

Since $G_{st}$ and $A_n$ are tightly coupled (see eq. 5), a damage of $A_n$ results in a simultaneous reduction in $G_{st}$ and $C_i$. The canopy-scale ozone flux into the leaf cavities ($F_{stC}$) is calculated by summing $F_{st}$ of all canopy layers, similar to the aggregation of $A_{n,l}$ and $G_{st}$ and $CUO_l$. Canopy $O_3$ concentration, ozone uptake, canopy cumulative ozone uptake (CUO) and damage to net photosynthesis are solved iteratively to account for the feedbacks between ozone damage, canopy conductance and canopy-air ozone concentrations.

The CUO above a threshold for trees and grass/crop PFTs together is referred to as $CUO_5^{1.6}$ in the following. Note that CUO and POD can be directly compared to estimates according to the (LRTAP-Convention, 2010) notation, when analysing only the top canopy layer (Mills et al., 2011b).





### 2.3 Sensitivity analysis

A sensitivity analysis is conducted to estimate the sensitivity of the modelled plant ozone uptake to the parameterisation of the model, to establish the robustness of the model, and to identify the most influential parameters. Three parameters ($\hat{R}_{gs}$, $r_{ext}$, $b$, see eq. 10, 6, 9, respectively) and two modelled quantities ($G_c$ and $R_a$, eq. 5, 2, respectively), with considerable uncertainty

due to the underlying parameters used to calculate these quantities, are perturbed within $\pm 20\%$ of their central estimate.

A set of 100 parameter combinations is created with a latin hypercube sampling method (McKay et al., 1979), simultaneously perturbing all five parameter values (R-package: FME, function: Latinhyper). For each parameter combination, a transient run (see Modelling protocol section) is performed creating an ensemble of estimates for the key prognostic variables $F_{stC}$ (eq. 13), $R_c$ (eq. 4), $V_g$ (eq. 2) and the ozone flux ratio ($F_R$) calculated as the ratio of $F_{stC}$ and the total ozone flux to the surface ($F_g$,

eq. 1).

The summer months June, July, and August (JJA) are selected from the simulation output and used for further analysis. For each prognostic variable ($F_{stC}$, $R_c$, $V_g$, $F_R$), the sensitivity to changes in all five perturbed parameters/variables is estimated by calculating partial correlation coefficients (PCC) and partial ranked correlation coefficients (PRCC) (Helton and Davis, 2002). PCC's record the linear relationship between two variables where the linear effects of all other variables in the analysis are

removed (Helton and Davis, 2002). In case of nonlinear relationships, RPCC can be used, which implies a rank transformation to linearise any monotonic relationship, such that the regression and correlation procedures as in the PCC can follow (Helton and Davis, 2002). We estimate the magnitude of the parameter effect by creating mean summer values of the four prognostic variables for each sensitivity run, and regressing these values against the corresponding parameter/variable scaling values of the respective model run.

### 2.4 Modelling protocol and data for site-level simulations

The site levels simulations at the FLUXNET sites are run using observed metrological forcing, soil properties, and land cover from the La Thuile data set of the FLUXNET project (Baldocchi et al., 2001). Data on atmospheric [$CO_2$] concentrations are obtained from (Sitch et al., 2015). Nitrogen deposition (reduced and oxidised deposition in wet and dry forms) and hourly ozone concentrations at 45 m height are provided by the EMEP model (see Sect. 2.5).

OCN is brought into equilibrium in terms of the terrestrial vegetation and soil carbon and nitrogen pools in a first step with the forcing of the year 1900. In the next step, the model is run with a progressive simulation of the period 1900 up untill the start year of the respective site. For this period atmospheric O3 and $CO_2$ concentrations, as well as N deposition of the respective simulated years are used. Due to lack of observed climate for the sites for this period, the site-specific observed meteorology from recent years is iterated for these first two steps. The observation years (see Appendix tab. 1) are simulated

with time-varying climate and atmospheric conditions (N deposition, $CO_2$ and $O_3$ concentrations).

For the evaluation of the model output, net ecosystem exchange (NEE), and latent heat flux (LE), as well as meteorological observations are obtained for eleven evergreen needle-leaved forest sites, ten deciduous broadleaved forest sites and five C3





grassland sites in Europe (see Appendix tab. 1) from the La Thuile data set of the FLUXNET project (Baldocchi et al., 2001). Leaf area indices (LAI) based on discrete point measurements are obtained from the La Thuille ancillary data base.

NEE measurements are used to estimate gross primary production (GPP) by the flux-partitioning method according to (Reichstein et al., 2005). Canopy conductance ($G_c$) is derived by inverting the Penmen-Monteith equation given the observed
LE and atmospheric conditions as described in (Knauer et al., 2015).

The half-hourly FLUXNET and model fluxes are filtered to reduce the effects of model biases on the model-data comparison. To derive average growing-season fluxes, night-time and morning/evening hours are excluded by removing data with lower than 20% of the daily maximum short-wave downward radiation. To avoid any biases associated with the soil moisture or atmospheric drought response of OCN, we further exclude data points with a modelled soil moisture constraint factor (range
between 0-1) below 0.8 and an atmospheric vapour pressure deficit larger than 0.5 kPa. Daily mean values are calculated of the remaining time steps where both modelled and observed values are present. The derived daily values are furthermore constrained to the main growing season by excluding days where the daily GPP is less than 20% of the yearly maximum daily GPP.

To derive representative diurnal cycles, data for the month July are filtered for daylight hours (taken as incoming short-wave
radiation $\geq 100$ W m$^{-2}$), and excluding periods of soil or atmospheric drought stress as above. This is done for modelled $F_{stC}$, $R_c$, $V_g$, $F_R$ and modelled as well as FLUXNET observed $GPP$ and $G_c$.

### 2.5 Modelling protocol and data for regional simulations

For the regional simulations, OCN is run at a spatial resolution of 0.5° x 0.5° on a spatial domain focused on Europe. Daily meteorological forcing (temperature, precipitation, short-wave and long-wave downward radiation, atmospheric specific hu-
midity and wind speed) for the years 1961 to 2010 is obtained from RCA3 regional climate model (Samuelsson et al., 2011; Kjellstrom et al., 2011), nested to the ECHAM5 model (Roeckner et al., 2006), and has been bias corrected for temperatures and precipitation using the CRU climatology (New et al., 1999). Nitrogen deposition (reduced and oxidised deposition in wet and dry forms) and ozone concentrations at 45 m height for the same years are obtained from the EMEP model, which is also run with RCA3 meteorology (as in Simpson et al., 2014b). Emissions for the EMEP runs in current years are as described
in (Simpson et al., 2014b), scaled back to 1900 using data from UN-ECE and van Aardenne et al. (2001) – see Appendix B. Further details of the EMEP model setup for this grid and meteorology can be found in Simpson et al. (2014b) and Engardt et al. (2016). For OCN, land cover, soil, and N fertiliser application are used as in (Zaehle et al., 2011) and kept at 2005 values throughout the simulation. Data on atmospheric [CO$_2$] concentrations are obtained from (Sitch et al., 2015).

OCN is brought into equilibrium in terms of the terrestrial vegetation and soil carbon and nitrogen pools with 1961-1970
forcing. This is followed by a simulation for the years 1960-2010 with time-varying climate and atmospheric conditions (N deposition, CO$_2$, and O$_3$ concentrations), but static land cover and land-use information (kept at year 2005 levels). An up-scaled FLUXNET-MTE-product of GPP (Jung et al., 2011) is used to evaluate modelled GPP.



## 2.6 Impacts of using the ozone deposition scheme

Different to other terrestrial biosphere models, the OCN ozone module accounts for the effects of aerodynamic, stomatal and non-stomatal resistance to ozone deposition. Due to these resistances the canopy O3 concentration is lower than the atmospheric $O_3$ concentration. Thus using such a deposition scheme reduces modelled ozone uptake into plants and accumulation.

5  To get an estimate of the magnitude of this impact we compare simulations with the standard deposition scheme as described above (D) with a simulation where ozone surface resistance is only determined by stomatal resistance and the non-stomatal depletion of ozone is zero (D-STO), and a further simulation where no deposition scheme is used and the canopy $O_3$ concentration is equal to the atmospheric concentration (ATM).

## 3 Results

10  ## 3.1 Evaluation against daily eddy-covariance data

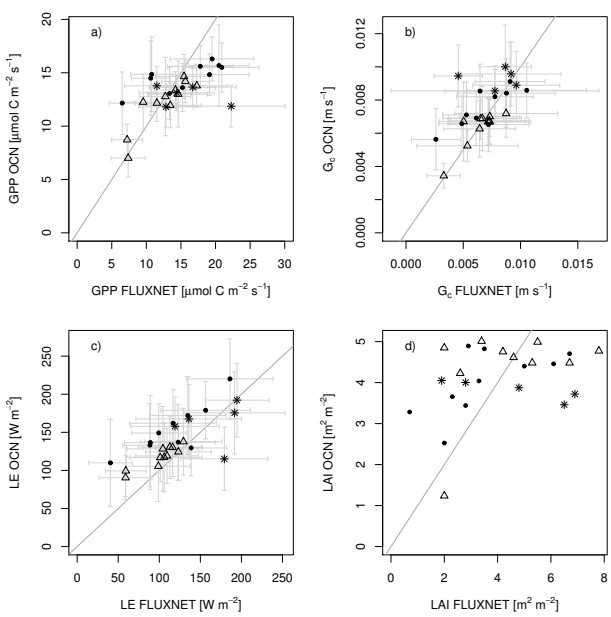

**Figure 1.** Comparison of measured a) GPP, b) canopy conductance ($G_c$), c) latent heat fluxes (LE) and d) LAI at 26 European FLUXNET sites and simulations by OCN. Displayed are means and standard deviations of daily means of the measuring/simulation period, with the exception of FLUXNET derived LAI, which is based on point measurements. Dots symbolise sites dominated by broadleaved trees, triangles sites dominated by needle-leaved trees and asterisks sites dominated by C3 grasses. The grey line constitutes the 1:1 line.

Figure 1 a shows that, for most sites, modelled and observation-based GPP agree within the standard deviation. The standard deviation is larger for the observation-based estimates because of the high level of noise in the eddy-covariance data. For sites dominated by needle-leaved trees, the modelled and observation-based GPP values are very close, with some slightly under-





and overestimates by the model at some sites. At sites dominated by broadleaved trees, modelled GPP deviates more strongly from the observation-based GPP, underestimating the observations in six out of ten cases. However, the results are within the range of standard deviation except for the drought prone PT-Mi1 site (see Appendix Fig. 10 a) for an explicit site comparison). At C3 grassland sites, modelled GPP is in good agreement with the observation-based GPP except for AT-Neu, which has

the highest mean GPP of all sites observed by FLUXNET with a arge standard deviation, which may reflect the effect of site management, not replicated by the model.

When comparing modelled and observed latent heat fluxes (LE), the model fits the observations best at the needle-leaved forest sites (Fig. 1 c). However, LE is overestimated at nine out of ten broadleaved forest sites, but remains within the range of the large observational standard deviation. At sites dominated by C3 grasses the modelled $LE$ differs considerably from observed

value, at two sites overestimating and two underestimating the fluxes, again within the observational-standard deviation.

In agreement with the comparison of GPP and LE, the comparison of modelled to observation-based canopy conductance ($G_c$) shows the best agreement for sites dominated by needle-leaved trees (Fig. 1 b). At sites dominated by broadleaved trees, the modelled $G_c$ varies more widely from the FLUXNET $G_c$. The modelled $G_c$ at sites dominated by C3 grasses is in very good agreement to FLUXNET $G_c$ except for the DE-Meh site, where means differ widely.

The comparison of the average modelled summertime LAI and point measurements at the FLUXNET illustrates that the variability in the measured LAI is much greater than that of OCN (Fig. 1 d). The modelled LAI values approach light-saturating, maximum LAI values and are not able to reproduce between-site differences in e.g. the growth stage, site-history, or maximum possible LAI values. Furthermore, it should be born in mind that the observed LAI values are averages of point measurements, which are not necessarily representative of the modelled time-period, and that the model had not been parameterised specifically

for the sites. The much better represented values of GPP, $G_c$ and LE compared to FLUXNET data (Fig. 1 a-c) indicate that OCN is able to adequately transform available energy into carbon uptake and water loss and thus to simulate key variables impacting ozone uptake within a reasonable range.

## 3.2 Mean diurnal cycles of key ozone parameters.

For further evaluation of the modelled ozone uptake, we analysed the diurnal cycles at three sites to represent typical ecosystems

of Europe. The selection criteria are that modelled and FLUXNET GPP and $LAI$ agree well and a minimum of five observation years is available to reduce possible biases from the inability of the model to simulate short-term variations from the mean. The selected sites are a temperate broadleaved summer green forest (IT-Ro1), a boreal needle-leaved evergreen forest (FI-Hyy), as well as a temperate C3 grass land (CH-Oe1). We evaluate modelled $GPP$ and $G_c$ against observations from the FLUXNET sites. We also present the modelled mean diurnal cycles of ozone related variables for which we did not have access to site-

specific observations ($F_{stC}$, $R_c$, $V_g$, $F_R$). Instead, we compare these quantities to reported values in the literature.

Modelled and observed mean diurnal cycles of $GPP$ and $G_c$ are in general agreement at the three selected FLUXNET sites (see Fig. 2 a,g,m) and b,h,n)) with a particular good agreement for the mean diurnal cycle of $GPP$ at the Finnish site FI-Hyy, where the hourly means are very close and the observational standard deviation is narrow (see Fig. 2 g). At the Italian site IT-Ro1 the overall daytime magnitude of the fluxes is reproduced in general except for the observed afternoon reduction in



$GPP$ (see Fig. 2 a). The hourly means are within the calculated standard deviation. Modelled and observation-based hourly means of $GPP$ at the site CH-Oe1 agree well except for the evening hours, where the observed values became highly variable. The mean diurnal cycles of $G_c$ derived from the FLUXNET data are again best matched at the site FI-Hyy, where as the model generally overestimates the diurnal cycle of $G_c$ slightly at the site IT-Ro1, and peak $G_c$ at the CH-Oe1 site. The fact that OCN

does not always simulated the observed midday depression of $G_c$, suggests that the response of stomata to atmospheric and soil drought in OCN requires further evaluation and improvement.

The modelled hourly mean ozone surface resistance $R_c$ is highest with approximately $400 \, \mathrm{sm^{-1}}$ during night time and decreases during daytime to values of $100\text{-}180 \, \mathrm{sm^{-1}}$, where the lowest surface resistance of approximately $100 \, \mathrm{sm^{-1}}$ is modelled at the grassland site CH-Oe1 (Fig. 2 f,l,r). These values are slightly higher than independent estimates (for grasses

and crops obtained for other sites) of noon surface resistances ranging $50\text{-}100 \, \mathrm{sm^{-1}}$ (Padro, 1996; Coyle et al., 2009; Gerosa et al., 2004; Tuovinen et al., 2004b). Tuovinen et al. (2004b) reported noon values of approximately $140 \, \mathrm{sm^{-1}}$ for a Scots pine forest and $70\text{-}140 \, \mathrm{sm^{-1}}$ for a Norway spruce forest site (Tuovinen et al., 2001), which compares well with the modelled $R_c$ values at the needle-leaved forest site (FI-Hyy; Fig. 2 l). Higher noon values of $250 \, \mathrm{sm^{-1}}$ are reported at a Danish Norway spruce site (Mikkelsen et al., 2004). For a Mountain Birch forest noon values of $110\text{-}140 \, \mathrm{sm^{-1}}$ (Tuovinen et al., 2001) are

observed what is slightly lower than the modelled value at the IT-Ro1 site (dominated by broadleaved tree PFT).

The modelled deposition velocities $V_g$ are lowest during night time with values of approximately $0.002 \, \mathrm{m\,s^{-1}}$ (Fig. 2 e,k,q). These values increase to maximum hourly means of $0.006\text{-}0.007 \, \mathrm{m\,s^{-1}}$ during daytime. These values compare well to reported values of deposition velocities, which range from $0.003\text{-}0.009 \, \mathrm{m\,s^{-1}}$ at noon (Gerosa et al., 2004) for a barely field, approximately $0.006 \, \mathrm{m\,s^{-1}}$ at noon for a wheat field (Tuovinen et al., 2004b) and approximately $0.009 \, \mathrm{m\,s^{-1}}$ at noon

at a potato field (Coyle et al., 2009). These estimates also agree well with maximum deposition velocities reported for Scots pine site of $0.006 \, \mathrm{m\,s^{-1}}$ (Keronen et al., 2003; Tuovinen et al., 2004b) and noon values from Danish Norway spruce sites of $0.006\text{-}0.010 \, \mathrm{m\,s^{-1}}$ (Mikkelsen et al., 2004; Tuovinen et al., 2001). Mean daytime deposition velocities of $0.006 \, \mathrm{m\,s^{-1}}$ (range $0.003 - 0.008 \, \mathrm{m\,s^{-1}}$) are reported at a finish mountain birch site (Tuovinen et al., 2001).

The stomatal ozone uptake $F_{stC}$ (Fig. 2 c,i,o) is close to zero during night time when the stomata are assumed to be

closed, because gross photosynthesis is zero. At FI-Hyy and CH-Oe1, peak uptake occurred at noon at values between $8\text{-}9$ $\mathrm{nmol\,m^{-2}\,s^{-1}}$, when photosynthesis (Fig. 2 g,m) and stomatal conductance (Fig. 2 h,n) are highest. At the Italian site IT-Ro1, maximum uptake occurs in the afternoon hours around $15 \, \mathrm{h}$, with much larger standard deviation compared to the other two sites (Fig. 2 c)). The magnitude of stomatal ozone uptake corresponds well to some values reported e.g. for crops (Gerosa et al., 2003, 2004, daily maxima of $4\text{-}9 \, \mathrm{nmol\,m^{-2}\,s^{-1}}$) and holm oak (Vitale et al., 2005, approx. $7\text{-}8 \, \mathrm{nmol\,m^{-2}\,s^{-1}}$). Lower daily

maximum values have been reported for an evergreen Mediterranean Forest dominated by Holm Oak of $4 \, \mathrm{nmol\,m^{-2}\,s^{-1}}$ under dry weather conditions (Gerosa et al., 2005) and $1\text{-}6 \, \mathrm{nmol\,m^{-2}\,s^{-1}}$ for diverse southern European vegetation types (Cieslik, 2004). Much higher values are reported for *Picea abies* ($50\text{-}90 \, \mathrm{nmol\,m^{-2}\,s^{-1}}$), *Pinus cembra* ($10\text{-}50 \, \mathrm{nmol\,m^{-2}\,s^{-1}}$) and *Larix decidua* ($10\text{-}40 \, \mathrm{nmol\,m^{-2}\,s^{-1}}$) at a site near Innsbruck Austria (Wieser et al., 2003), where canopy ozone uptake was estimated by sapflow measurements in contrast to the studies mentioned before where the eddy covariance technique was



applied. The much higher $F_{stC}$ values in that study result from much higher canopy conductances to ozone ($G_c^{O_3}$), which are up to 12 times higher than the modelled $G_c^{O_3}$ values in our study (see Fig. 2, $\frac{G_c}{1.51}$).

The ratio between the vegetation ozone uptake and the total surface uptake ($F_R$) is close to zero during night time hours and increases steeply in the morning hours (Fig. 2 d,j,p). The 24 h average is approximately 0.3 for IT-Ro1 and 0.4 for FI-Hyy and

CH-Oe1 (Fig. 2 d,j,p). Peak hourly mean values are close to 0.6 at IT-Ro1, around 0.7 at FI-Hyy and close to 0.8 at CH-Oe1. These values are comparable to the ratios reported for crops (0.5 - 0.6 Gerosa et al., 2004; Fowler et al., 2009), Norway spruce (Mikkelsen et al., 2004, 0.3 -0.33) and diverse southern European vegetation types (Cieslik, 2004, 0.12 - 0.69). The modelled flux ratios here show slightly higher daily maximum flux ratios than reported in the listed studies. Daily mean flux ratios are well within the reported range.

## 3.3 Sensitivity analysis

We assess the sensitivity of the modelled ozone uptake and deposition, represented by $F_g$, $F_{stC}$, $V_g$ and $R_c$ to uncertainty in five weakly constraint variables and parameters of the ozone deposition scheme ($R_a$, $b$, $r_{ext}$, $\hat{R}_{gs}$ and $G_c$). Fig. 3 a shows for example the results for the Finnish needle-leaved forest FI-Hyy. As expected, all uptake/deposition variables, except of the flux ratio $F_R$ are negatively correlated to the aerodynamic resistance $R_a$, describing the level of decoupling of the atmosphere and

land surface. Increasing $R_a$ decreases the canopy internal $O_3$ concentration and hence stomatal ($F_{stC}$) and total ($F_g$) deposition as well as the deposition velocity ($V_g$). The flux ratio $F_R$ is slightly positively correlated to changes in $R_a$ due to the stronger negative correlation of $F_{stC}$ compared to $F_g$.

In decreasing order, but as expected, the level of external leaf-resistance ($r_{ext}$), the scaling factor $b$ (eq. 9), and soil resistance ($\hat{R}_{gs}$) increase $R_c$ and consequently reduce $F_g$ and $V_g$. Reducing the non-stomatal deposition by increasing $r_{ext}$, $b$, and $\hat{R}_{gs}$

increases the canopy internal $O_3$ concentration and thus stomatal ozone uptake ($F_{stC}$). The combined effects of a reduction of total deposition $F_g$ and an increase of $F_{stC}$ cause a positive correlation of $F_R$ to $r_{ext}$, $b$, and $\hat{R}_{gs}$.

Increasing $G_c$ increases stomatal ozone uptake ($F_{stC}$) and thereby also increases $V_g$ and $F_g$. The increased total ozone uptake ($F_g$) decreases the surface resistance to ozone uptake $R_c$ what causes a negative correlation of $R_c$ with $G_c$. The stronger increase in $F_{stC}$ compared to $F_g$ results in a positive correlation of $F_R$.

Despite these partial correlations, only changed values for $r_{ext}$ and $G_c$ have a notable effect on the predicted fluxes (Fig. 3 b), whereas for the other factors ($R_a$, $b$, and $\hat{R}_{gs}$), the impact on the simulated fluxes is less than 0.1 (% / %). The flux ratio $F_R$ is very little affected by varying $r_{ext}$ and $G_c$.

Notwithstanding the perturbations, all four ozone related flux variables show a fairly narrow range of simulated values (Fig. 4). The seasonal course of the surface resistances and fluxes are maintained. The simulations show a strong day to day

variability of $F_{stC}$, which is conserved with different parameter combinations, and which is largely driven by the day-to-day variations in $G_c$ and the atmospheric ozone concentration $O_3$ (see Fig. 4 f and e respectively). Ozone uptake by the leaves reduces the ozone surface resistance during the growing season such that $R_c$ becomes lowest. The cumulative uptake of ozone (CUO) is lowest at the beginning of the growing season but not zero because the evergreen pine at the Hyytiälä site accumulates



ozone over several years (Fig. 4 f). The CUO increases during the growing season and declines in autumn when a larger fraction of old needles are shed.

### 3.4 Regional simulations

We used the model to simulate the vegetation productivity, ozone uptake, and associated ozone damage of plant production
over Europe for the period 2001-2010 (see Section 2.5 for modelling protocol).

Simulated mean annual GPP for the years 1982-2011 shows in general good agreement with an independent estimate of GPP based on up scaled eddy-covariance measurements (MTE, see Section 2.5), with the estimates being within $250 \, \mathrm{g \, C \, m^{-2} \, yr^{-1}}$. A significant exception to this acceptable agreement are cropland dominated areas (Fig. 5) in parts of Eastern Europe, Southern Russia, Turkey and Northern Spain, which show consistent overestimation of GPP by OCN of 400 to $900 \, \mathrm{g \, C \, m^{-2} \, yr^{-1}}$.
Regions with a strong disagreement coincide with high simulated LAI values by OCN and a higher simulated GPP in summer compared to the summer GPP by MTE. In addition, OCN simulates a longer growing season for croplands since sowing and harvest dates are not considered. It is worth noting, nevertheless, that there are no FLUXNET stations present in the regions of disagreement hotspots, making it difficult to assess the reliability of the MTE product in this region.

North of $60°N$ OCN has the tendency to produce larger estimates of GPP than inferred from the observation-based product,
which is particularly pronounced in low productivity mountain regions of Norway and Sweden. It is unclear whether this bias is indicative of a too strong N limitation in the OCN model.

Average decadal ozone concentrations generally increase from Northern to Southern Europe (Fig. 6 a) and with increasing altitude, with local deviations from this pattern in centres of substantial air pollution. The pattern of foliar ozone uptake differs distinctly from that of the O3 concentrations, showing highest uptake rates in Central, Eastern and parts of Southern Europe
(Fig. 6 b), associated with centres of high rates of simulated gross primary production (Fig. 6 d) and thus canopy conductance. The cumulative ozone uptake beyond the PFT-specific detoxification threshold reaches values of 6-12 $\mathrm{mmol \, m^{-2}}$ in large parts of Central Europe (Fig. 6 c). The highest accumulation rates of up to 15-20 $\mathrm{mmol \, m^{-2}}$ are found in temperate Eastern Europe between $50°N$ and $60°N$, in Italy, the Alps and the Bordeaux region. The concentration based exposure index AOT40 (Fig. 6 d) shows a strong north south gradient similar to the ozone concentration (Fig. 6 a) and is distinctly different to the flux based
$CUO_5^{1.6}$ pattern (Fig. 6 c).

Simulated reduction of mean decadal GPP due to ozone averaged 60-120 $\mathrm{g \, C \, m^{-2} \, yr^{-1}}$ over large areas of Central, Eastern, and South-eastern Europe and is generally largest in regions of high productivity (fig 7 a). The relative reduction of GPP is fairly consistent across larger areas in Europe and averages 4-6%. Higher reductions in relative terms are found in regions with high cover fraction of C4 PFT's (see Appendix 12 a,b) like in the Black see area. Lower relative reductions are found in
Northern and parts of Southern Europe where productivity is low and probably either low $O_3$ concentrations or drought control on stomatal fluxes respectively reduces stomatal $O_3$ uptake. Slight increases or strong decreases in relative terms are found in regions with very small productivity like in Northern Africa. A slight increases in GPP might be caused by feedbacks of GPP damage on LAI, canopy conductance and soil moisture content such that e.g. water savings enable a prolonged growing





season and thus a slightly higher GPP. Overall, simulated European productivity has been reduced from $10.6 \, \mathrm{Pg\,C\,yr^{-1}}$ to $10.1$ $\mathrm{Pg\,C\,yr^{-1}}$ corresponding to a 4.7% reduction.

The ozone induced reductions in GPP are associated with a reduction in mean decadal transpiration rates by $5\text{-}10 \, \mathrm{mm\,yr^{-1}}$ over large parts of Central and Eastern Europe (7 c). These reductions correspond to 3-4% in Central European and to 4-6%

in Northern Europe. As expected, the reductions in transpiration rates are therefore slightly less than for GPP due to the role of aerodynamic resistance in controlling water fluxes in addition to canopy conductance. Very high reductions in transpiration are found in the Eastern Black see area associated with strong reductions in GPP. Regionally (in particular in Eastern Spain, Northern Africa and around the Black Sea) lower reductions in transpiration or even slight increases are found (7 d). They are related to ozone induced soil moisture savings during the wet growing season, leading to lower water stress rates during the

drier season. Overall, simulated European mean transpiration has been reduced from $170.4 \, \mathrm{mm}$ to $165.7 \, \mathrm{mm}$ corresponding to a 2.8% reduction.

### 3.5 Impacts of using the ozone deposition scheme

At the FI-Hyy site the canopy O3 concentration, uptake and accumulation (CUO) increases approximately 10-15% for the D-STO model and 20-25% for the ATM model version compared to the standard deposition scheme (D) used here (Fig. 8a-c

and Appendix 11). The $\mathrm{CUO}_{1.6}$ increases stronger and constitutes 35% and 65% for the D-STO and ATM model, respectively. The exact values however are site and PFT specific (see Appendix 11 for the CH-Oe1 and IT-Ro1 site).

The regional impact of using the ozone deposition scheme on CUO is shown in Fig. 9 (for $\mathrm{CUO}_5^{1.6}$ see Appendix 13). CUO substantially decreases for the D-STO (Fig. 9b) compared to the ATM model (Fig. 9a). Using the standard deposition model D (Fig. 9c) further reduces the CUO compared to the ATM version where the stomata respond directly to the atmospheric ozone

concentration.

Calculating the canopy ozone concentration with the help of a deposition scheme that accounts for stomatal and non-stomatal ozone deposition thus reduces ozone accumulation in the vegetation.

## 4   Discussion

We extended the terrestrial biosphere model OCN by a scheme to account for the atmosphere–leaf transfer of ozone with the

aim to better account for air pollution effects to net photosynthesis and hence the regional to global water, carbon, and nitrogen cycling. This ozone deposition scheme calculates canopy $O_3$ concentrations and uptake into the leaves depending on surface conditions and vegetation carbon uptake. We show that using the canopy $O_3$ concentration strongly impacts the cumulative uptake of ozone (CUO) and $\mathrm{CUO}_5^{1.6}$ compared to assuming that the $O_3$ concentration outside the leaf would be identical to the atmospheric $O_3$ concentration in $45 \, \mathrm{m}$ height as provided by the CTM. Perturbations of key variables and parameters

of the implemented ozone deposition scheme show little impact on the simulated ozone uptake and deposition variables. In other words, the calculated ozone uptake is relatively robust against uncertainties in the parameterisation of some of the lesser known surface properties. Our sensitivity analysis shows that a further crucial part for calculating plant ozone uptake is a




correct estimate of the canopy conductance. We provide an assessment of the modelled canopy conductance, and find that the model produces reasonable estimates of canopy conductance compared to FLUXNET data, with a range of caveats as discussed in Section 4.2. We relate accumulated ozone uptake above a PFT-specific threshold to reductions in net photosynthesis and find that across large regions of Europe, ozone reduces production and transpiration by approximately 5% and 3%, respectively.

This reflects the shape of the implemented damage function which is further discussed in Section 4.3.

## 4.1 Atmosphere-leaf transport

A crucial component for calculating canopy ozone uptake $F_{stC}$ – besides a reliable estimate of $G_c$ – is a reliable estimate of surface ozone concentrations. Ozone destruction above and within the canopy airspace due to compounds emitted by the plants (e.g. biological volatile organic compounds, BVOC's) is assumed to be (at least partly) implicitly included into the

non-stomatal ozone destruction terms included in both the EMEP CTM and OCN deposition frameworks. To evaluate the functionality of the implemented ozone deposition scheme in OCN, mean simulated diurnal cycles of key ozone deposition and uptake variables are calculated and found to be within the range of values reported in the literature (see Section 3.2). The implemented deposition scheme is therefore assumed to produce realistic values for key variables.

Analysing partial correlation coefficients and the strength of the correlation calculated from the sensitivity runs shows that

the $F_{stC}$ is most sensitive to changes in $G_c$. This emphasise the importance of reliable estimates of canopy conductances to obtain reliable estimates of ozone uptake.

## 4.2 Site-level evaluation

Our results indicates the importance of reliable estimates of the canopy conductance ($G_c$) for the calculation of ozone uptake. The site-level evaluation of simulated gross primary productivity (GPP), canopy conductance, and latent heat fluxes (LE) to

FLUXNET observations at 26 European sites across diverse ecosystem types shows a general good model-data agreement.

Eddy covariance measurements and derived flux and conductance estimates are subject to a diverse set of random and systematic errors (Richardson et al., 2012). A lack of energy balance closure can cause underestimation of sensible and latent heat as well as an overestimation of available energy, with a mean bias of 20 % where the imbalance is greatest during nocturnal periods (Wilson et al., 2002). This imbalance propagates to estimates of canopy conductance, which is inferred from latent and

sensible heat fluxes. The energy imbalance furthermore appears to affect estimates of $CO_2$ uptake and respiration (Wilson et al., 2002). Flux partitioning algorithms which extrapolate night-time ecosystem respiration estimates to daytime impose an additional potential for bias in the estimation of GPP (Reichstein et al., 2005).

The good agreement of seasonal mean $G_c$ at most of the 26 FLUXNET sites and the well reproduced diurnal cycles at the three selected sites indicates that the physiological processes simulated by OCN are suitable to replicate observed patterns of

$G_c$. This finding, together with the finding that modelled values of $V_g$, $R_c$ and $F_R$ are within observed ranges, encourages the use of the extended OCN model for determining the effect of air pollution on terrestrial carbon, nitrogen, and water cycling.



### 4.3 Regional damage estimates

The regional damage estimates of annual average GPP (- 4.7%) and T (- 2.8% ) simulated by OCN for the period of the years 2001-2010 are lower compared to previous reported estimates. Meta-analysis suggest on average a 11% (Wittig et al., 2007) and a 21% (Lombardozzi et al., 2013) reduction of instantaneous photosynthetic rates. However because of carry-over

effects this does not necessarily translate directly into reductions in annual GPP. Damage estimates using the Community land model (CLM) suggest GPP reductions of 10-25% in Europe and 10.8% globally (Lombardozzi et al., 2015). Reductions in transpiration are estimated 5-20% for Europe and globally 2.2% (Lombardozzi et al., 2015). Lombardozzi et al. (2015) however used fixed reductions of photosynthesis (12-20%) independent of cumulative ozone uptake for 2 out of 3 simulated plant types. Only for one plant type damage was related to cumulative ozone uptake with a very small slope. Sitch et al. (2007)

simulated global GPP reductions of 8-14% (under elevated and fixed $CO_2$ respectively) for low plant ozone sensitivity and 15-23% (under elevated and fixed $CO_2$ respectively) for high plant ozone sensitivity for the year 2100 compared to 1901. For the Euro-Mediterranean-region an average GPP reduction of 22% was estimated by the ORCHIDEE-model for the year 2002 using an AOT40 based approach (Anav et al., 2011).

  Possible causes for the deviations are the usage of very different dose-response-relationships, different flux thresholds ac-

counting for the detoxification ability of the plants, differing atmospheric ozone concentrations, non-identical simulation periods, and differences in simulating climate change (elevated $CO_2$) and air pollution (nitrogen deposition).

  An important factor in the difference to the previous study is the use of the ozone deposition scheme included in our study, which reduces $O_3$ surface concentrations, and hence also the estimated ozone uptake and accumulation (see fig. 9). Compared to the values that would have been obtained if the CTM $O_3$ concentrations of the atmosphere (from ca. 45 m height) had been

used directly at the leaf surface, our simulations yield a decrease of CUO by 31% ($CUO_5^{1.6}$ 65%) (European means for the years 2001-2010). A significant fraction of the decreases is associated with the non-stomatal ozone uptake and destruction at the surface, which decreased the simulated cumulative ozone uptake by 16% ($CUO_5^{1.6}$ 39%). To obtain an accurate as possible estimate of CUO/ $CUO_5^{1.6}$, stomatal and non-stomatal destruction of ozone and it's impacts on canopy ozone concentrations should be considered in terrestrial biosphere models (Tuovinen et al., 2009b).

The use of a flux threshold (possibly PFT specific) and it's magnitude naturally also impacts the CUOY (canopy cumulative ozone uptake above a threshold of Y $\mathrm{nmol\,m^{-2}\,s^{-1}}$) and possible damage estimates (Tuovinen et al., 2007). The mean decadal CUO for Europe in the years 2001-2010 is 43.6 $\mathrm{mmol\,m^{-2}}$ in our simulations whereas the mean $CUO_5^{1.6}$ is only 4.7 $\mathrm{mmol\,m^{-2}}$. Recent studies suggest flux thresholds of 6 $\mathrm{nmol\,m^{-2}\,s^{-1}}$ for crops and 0 or 1 $\mathrm{nmol\,m^{-2}\,s^{-1}}$ for trees and semi natural vegetation (LRTAP-Convention, 2010; Mills et al., 2011b). The impacts of using different flux thresholds on regional

estimates of ozone uptake, accumulation and damage are still poorly understood and need further research.

  A key aspect of ozone damage estimates are the assumed dose-response-relationships, which relate ozone uptake to plant damage. Several dose-response-relationships exist for biomass or yield damage (LRTAP-Convention, 2010, for an overview), but only few estimates exists for the likely cause of this damage, i.e. the reduction in net photosynthesis. In this study, the damage relationship to net photosynthesis proposed by Wittig et al. (2007) is used. The major advantage of this relationship




is that it has been obtained by meta-analysis of many different tree species and thus might indicate an average response. This relationship is therefore used for all modelled plant functional types. However, a substantial disadvantage is that the meta-analysis implies a damage of 6.16 % at zero accumulated ozone uptake with a rather minor increase in damage with increasing ozone uptake.

The use of flux-based damage estimates is generally thought to improve damage estimates compared to concentration based estimates (e.g. AOT40), since stomatal constraints on ozone uptake are taken into account, yielding very different spatial patterns of exposure hot spots (Simpson et al., 2007a). Similar to Simpson et al. (2007a), we find strongly differing patterns between cumulative ozone uptake above a threshold ($CUO_5^{1.6}$) and AOT40 in our simulations here (see Fig. 6), where highest exposure is not only found in southern Europe where the ozone concentration is highest but also in eastern Europe.

To elucidate the reasons for the substantial differences in the damage estimates further studies are necessary to disentangle the combined effects of differing flux thresholds, damage relationships, climate change, and nitrogen deposition which acts as a ozone precursor on the one hand and a growth enhancing nutrient on the other hand. Decoupling of photosynthesis and stomatal conductance might also impact GPP and transpiration damage estimates and should be regarded too. Accounting for direct impairment of the stomata might reduce the reported reductions in transpiration or even cause an increase compared to

simulations with no ozone damage.

## 5   Conclusion

Estimates of ozone impacts on plant gross primary productivity vary substantially. This uncertainty in the magnitude of damage and hence the potential impact on the global carbon budget is related to different approaches to model ozone damage. The use of a comparatively detailed ozone deposition scheme that accounts for non-stomatal as well as stomatal deposition, when

calculating surface $O_3$ concentrations substantially impacts ozone uptake in our model. We therefore recommend to generally consider non-stomatal ozone uptake in models estimating ozone damage to obtain a better estimate of ozone uptake and accumulation. We show that ozone uptake into the stomata is mainly impacted by the canopy conductance in the used ozone deposition scheme. This highlights the importance of modelling reliable canopy conductances besides realistic surface $O_3$ concentrations to obtain accurate as possible estimates of ozone uptake which are the basis for plant damage estimates. Suitable

ozone damage relationships to net photosynthesis for different plant groups are essential to relate the accumulated ozone uptake to plant damage in a model. Desirable are mean responses of plant groups similar to commonly modelled plant functional types. Only few relationships exist which indicate mean responses of several species e.g. Wittig et al. (2007) and Lombardozzi et al. (2013) which however propose very different relationships. Furthermore, the impact of the plants ability to detoxify ozone should be regarded e.g. by using flux thresholds as well as the combined effects of ozone with air pollution (nitrogen

deposition) and climate change (elevated $CO_2$) on the plants carbon uptake.



## Appendix A: aerodynamic resistance(Appendix material)

To calculate the ozone deposition of the free atmosphere at the lowest level of the CTM (approximately 45 m) to the vegetation canopy, it is necessary to know the aerodynamic resistance between these heights. ($R_{a,45}$). This data is model and land-cover specific, and thus not provided by the CTM. Instead, we approximate $R_{a,45}$ from the wind speed in 45 m height ($u_{45}$) and the

friction velocity $u_*$ according to

$$R_{a,45} = \frac{u_{45}}{u_*^2} \tag{A1}$$

where $u_*$ is calculated from $u_{10}$ using the atmospheric resistance calculations of the ORCHIDEE model (Krinner et al., 2005). The wind at 45 m ($u_{45}$) is approximated by assuming the logarithmic wind profile for neutral atmospheric conditions (Monteith and Unsworth, 2007) due to the lack of information on any other relevant atmospheric properties in 45 m height:

$$
\begin{array}{ll}
& u_{45} = u_{10} \dfrac{log(\frac{45}{z_0})}{log(\frac{10}{z_0})} \tag{A2}
\end{array}
$$

where $z_0$ is the roughness length.

## Appendix B: Emissions inventory

Emissions for the EMEP model were derived by merging data from three main sources. Firstly, emissions for 2005 and 2010 were taken from the so-called ECLIPSE database produced by IIASA for various EU Projects and the Task Force on Hemi-

spheric Transport of Air Pollution (Amann et al., 2013; Stohl et al., 2015), although with improved spatial resolution over Europe by making use of the 7 km resolution MACC-2 emissions produced by TNO (Kuenen et al., 2011). For 1990, emissions from land-based sources were taken directly from the EMEP database for that year, since 1990 had been the subject of recent review and quality-control (e.g. Mareckova et al., 2013). Emissions between 1990 and 2005 were estimated via linear interpolation between these 2005 and EMEP 1990 values. Emissions prior to 1990 were derived by scaling the EMEP 1990

emissions by the emissions ratios found in the historical data-series of Lamarque et al. (2010).

Emissions of the biogenic hydrocarbon isoprene from vegetation are calculated using the model's land cover and meteorological data (Simpson et al., 2012b, 1999). Emissions of NO from biogenic sources (NO from soils, forest-fires, etc) were set to zero given both their uncertainty and sporadic occurance. Tests have shown that this approximation has only a small impact on annual deposition totals to the EU area, even for simulations at the start of the 20th century. Volcanic emissions were set to

a constant value from the year 2010.



*Acknowledgements.* We would like to thank Magnuz Engardt of the Swedish Meteorological and Hydrological Institute for providing the RCA3 climate dataset. This research leading to this publication was supported by the EU Framework programme through grant no. 282910 (ECLAIRE), and the Max Planck Society for the Advancement of Science e.V. through the ENIGMA project. This project has received funding from the European Research Council (ERC) under the European Union's Horizon 2020 research and innovation programme (grant

5   agreement no. 647204; QUINCY).




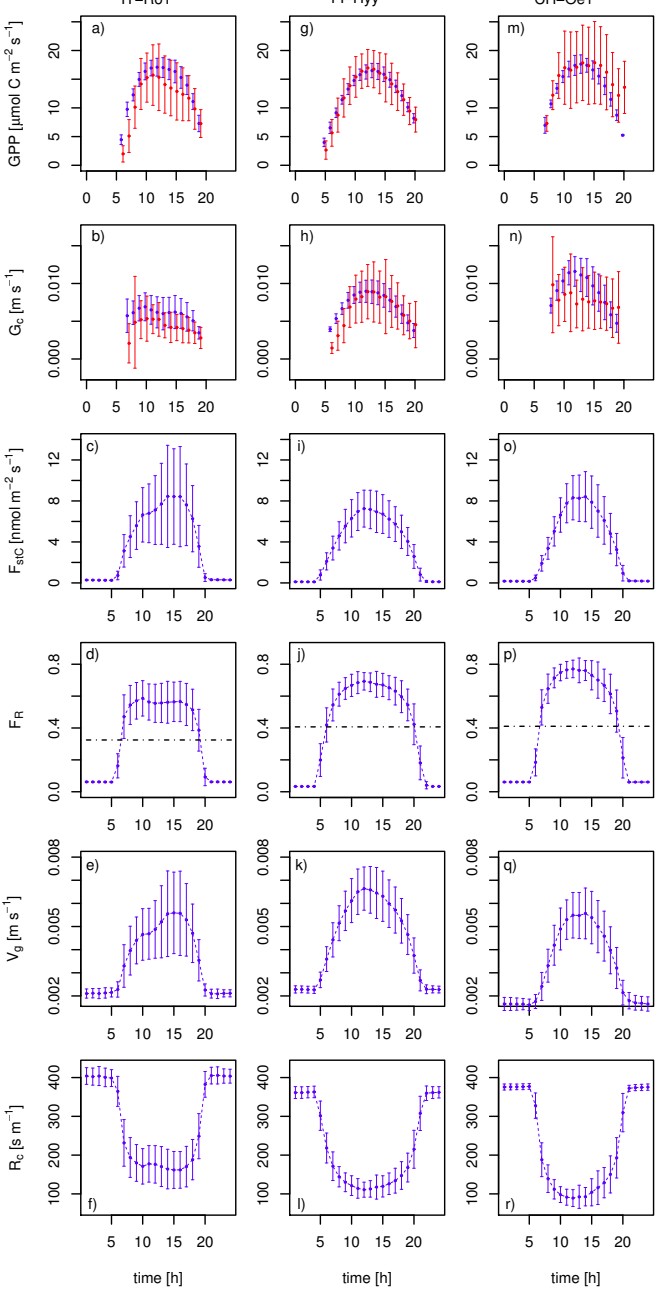

**Figure 2.** Simulated and observed hourly means over all days of the July months 2002 - 2006 for CH-Oe1 and IT-Ro1, and for 2001 - 2006 for FI-Hyy. Plotted are mean hourly values of a,g,m) GPP (blue: OCN, red: FLUXNET), b,h,n) canopy conductance ($G_c$) (blue: OCN, red: FLUXNET), c,i,o) ozone uptake ($F_{stC}$), d,j,p) the flux ratio $F_R$, e,k,q) ozone deposition velocity ($V_g$) and f,l,r) ozone surface resistance ($R_c$). The error bars indicate the standard deviation from the hourly mean. The dotted line in d,j,p) indicates the daily mean value.





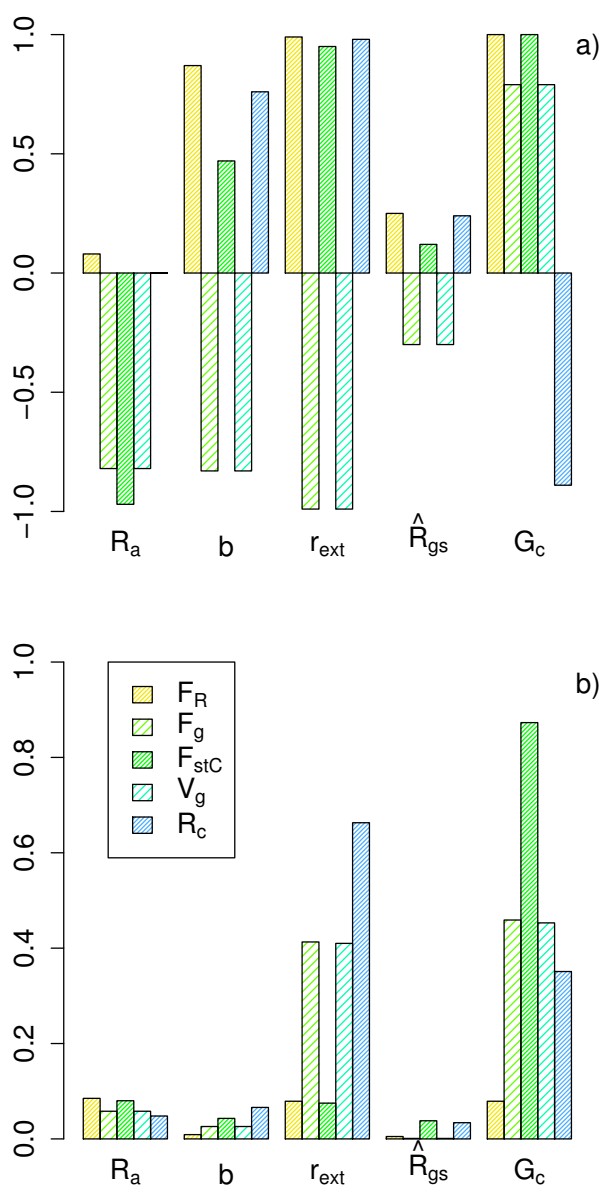

**Figure 3.** a) Mean partial correlation coefficients and b) strength of the correlation in % per %. $R_a$, $b$, $r_{ext}$, $\hat{R}_{gs}$ and $G_c$ are perturbed within $\pm 20\%$ of their central estimate. Results from simulations at the FLUXNET site FI-Hyy for the simulation period 2001-2006.





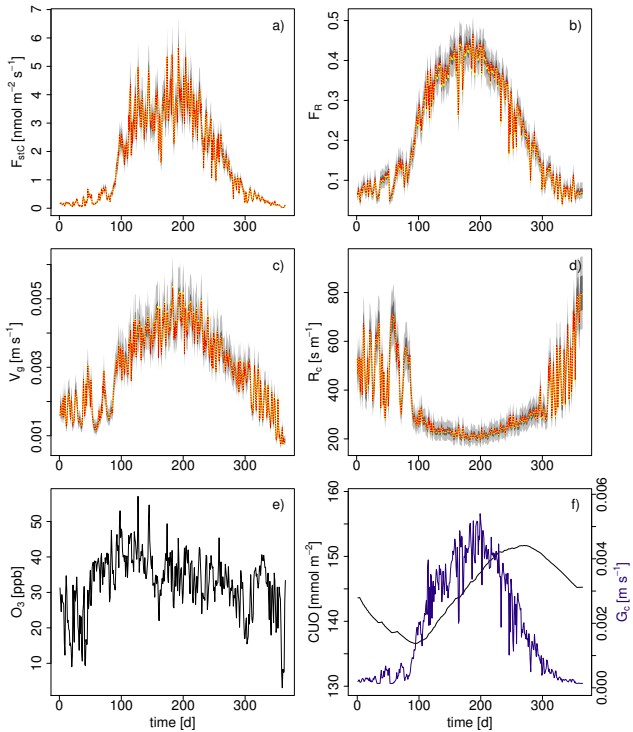

**Figure 4.** Ensemble range of key ozone uptake/deposition variables resulting from the perturbation of $R_a$, $b$, $r_{ext}$, $\hat{R}_{gs}$ and $G_c$ within $\pm 20\%$ of their central estimate. Shown are simulated daily mean values of a) ozone uptake ($F_{stC}$), b) the ozone flux ratio ($F_R$), c) ozone deposition velocity ($v_g$) and d) ozone surface resistance ($R_c$) for the boreal needle-leaved evergreen forest at the finish FLUXNET site FI-Hyy for the year 2001. Red dashed: unperturbed model; yellow: median of all sensitivity runs; dark grey area: interquartile-range; light grey area: min-max-range off all sensitivity runs. Simulated daily mean values for the respective site and year of e) atmospheric ozone concentrations $O_3$ and f) cumulative uptake of ozone (CUO) and canopy conductance $G_c$.

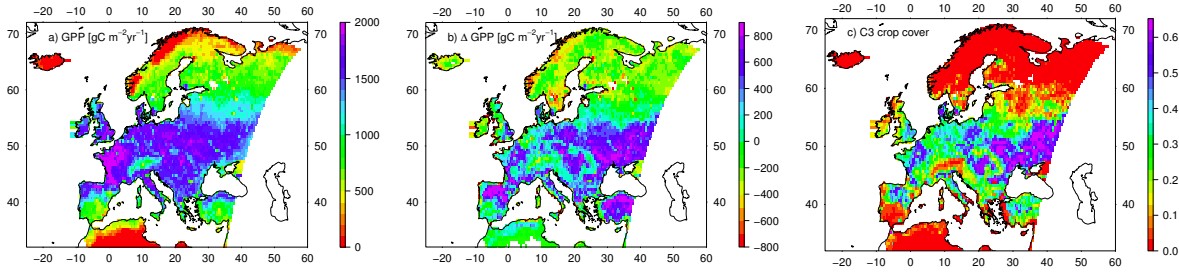

**Figure 5.** Europe-wide simulated GPP and difference between modelled GPP by OCN and a GPP estimate by a FLUXNET-MTE-product. Plotted are for the years 1982-2011 a) the simulated mean GPP accounting for ozone damage in $\mathrm{g\,C\,m^{-2}\,yr^{-1}}$, b) the mean differences for OCN - MTE GPP in $\mathrm{g\,C\,m^{-2}\,yr^{-1}}$ and c) the mean simulated grid cell cover of the C3-crop PFT in OCN, given as fractions of the total grid cell area.



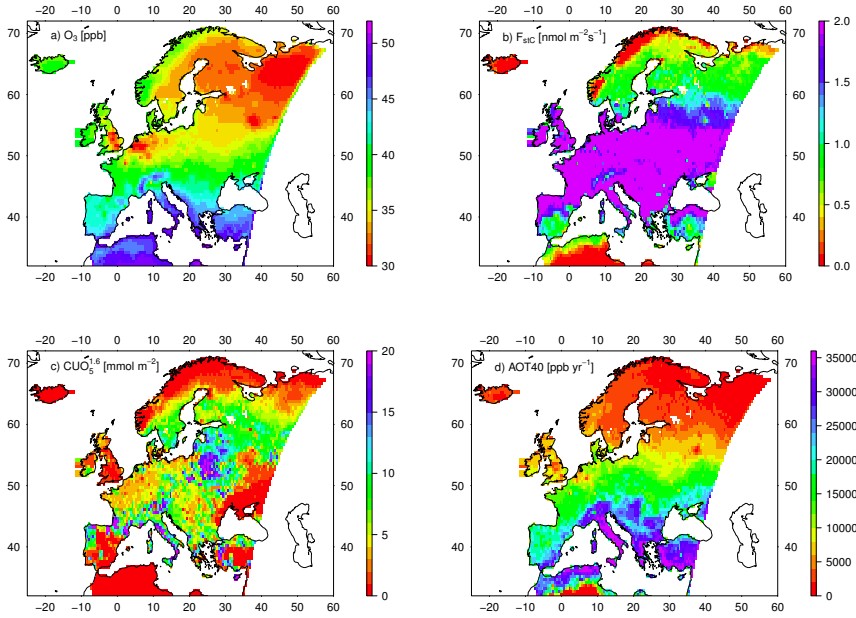

**Figure 6.** Mean decadal a) ozone concentration [ppb], b) canopy integrated ozone uptake into the leafs [nmol m$^{-2}$ s$^{-1}$], c) canopy integrated cumulative uptake of ozone above a threshold (CUO$_5^{1.6}$) [mmol m$^{-2}$] and d) AOT40 [ppm yr$^{-1}$], for Europe of the years 2001-2010.

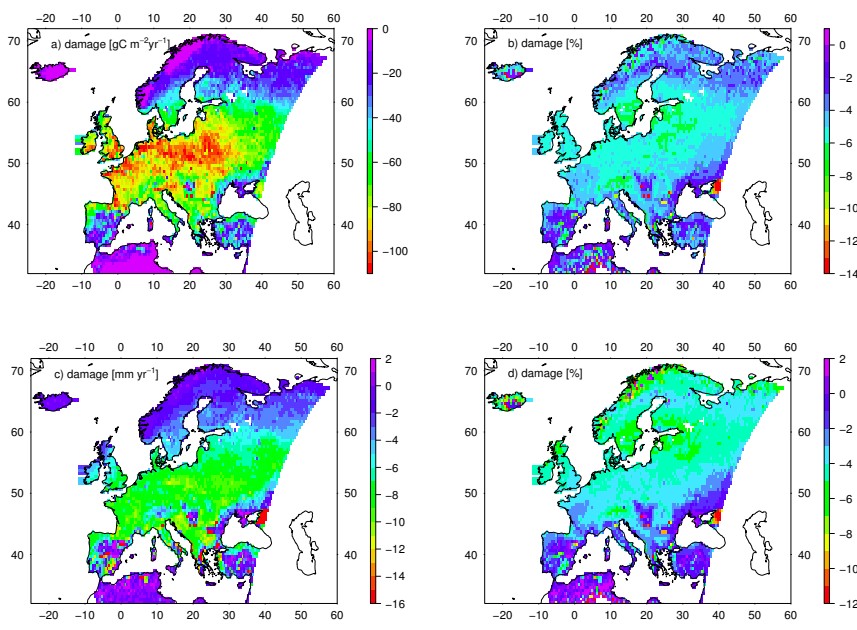

**Figure 7.** Mean decadal a) reduction in GPP [g C m$^{-2}$ yr$^{-1}$], b) percent reduction in GPP, c) reduction in transpiration [mm yr$^{-1}$] and d) percent reduction in transpiration due to ozone damage averaged for the years 2001-2010.




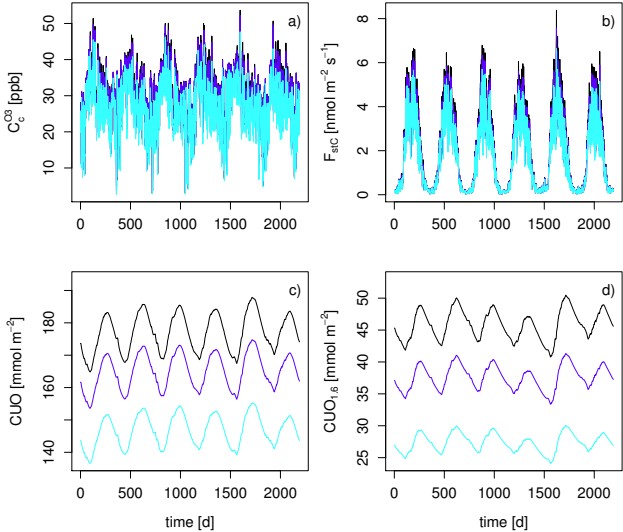

**Figure 8.** Mean daily values of the a) ozone surface concentration [ppb], b) canopy integrated ozone uptake into the leafs [$\mathrm{nmol\,m^{-2}\,s^{-1}}$], c) canopy integrated cumulative uptake of ozone (CUO) [$\mathrm{mmol\,m^{-2}}$] and d) canopy integrated cumulative uptake of ozone above a threshold (CUO$_{1.6}$) [$\mathrm{mmol\,m^{-2}}$] at the FLUXNET site FI-Hyy. Black: ATM model, Dark blue: D-STO model, Light blue: standard deposition model (D).

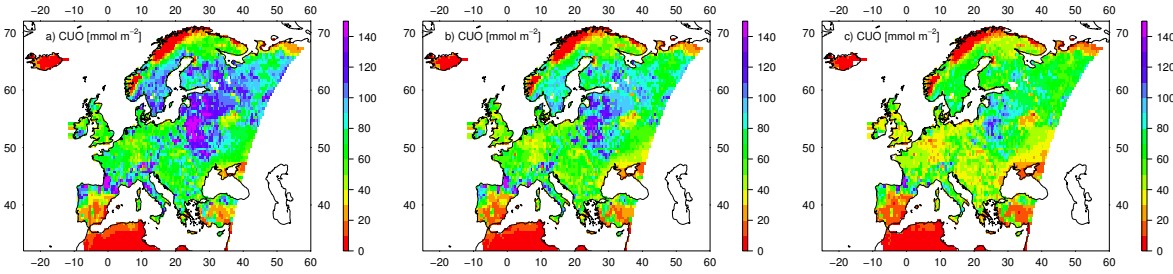

**Figure 9.** Mean decadal canopy integrated cumulative uptake of ozone (CUO) [$\mathrm{mmol\,m^{-2}}$] for Europe of the years 2001-2010. a) no ozone deposition scheme (ATM), b) ozone surface resistance is only determined by stomatal resistance (D-STO) and c) standard ozone deposition scheme (D).




**Table 1.** Characteristics of the FLUXNET sites used in this study.

|   | Site | Latitude | Longitude | Climate [a] | PFT [b] | Years | Reference |
|---|------|----------|-----------|-------------|---------|-------|-----------|
| 1 | AT-Neu | 47.12 | 11.32 | Cfb | TeH | 2002- 2005 | (Wohlfahrt et al., 2008b) |
| 2 | CH-Oe1 | 47.29 | 7.73 | Cfb | TeH | 2002- 2006 | (Ammann et al., 2007) |
| 3 | DE-Bay | 50.14 | 11.87 | Cfb | CEF | 1997- 1998 | (Rebmann et al., 2004) |
| 4 | DE-Hai | 51.08 | 10.45 | Cfb | TeBDF | 2000- 2006 | (Kutsch et al., 2008) |
| 5 | DE-Meh | 51.28 | 10.66 | Cfb | TeH | 2004- 2006 | (Scherer-Lorenzen et al., 2007) |
| 6 | DE-Tha | 50.96 | 13.57 | Cfb | CEF | 2004- 2006 | (Grünwald and Bernhofer, 2007) |
| 7 | DK-Lva | 55.68 | 12.08 | Cfb | TeH | 2005- 2006 | (Gilmanov et al., 2007) |
| 8 | DK-Sor | 55.49 | 11.65 | Cfb | TeBDF | 1997- 2006 | (Lagergren et al., 2008) |
| 9 | ES-ES1 | 39.35 | -0.32 | Csa | CEF | 1999- 2004 | (Sanz et al., 2004) |
| 10 | FI-Hyy | 61.85 | 24.29 | Dfc | CEF | 2001- 2006 | (Suni et al., 2003) |
| 11 | FR-Hes | 48.67 | 7.06 | Cfb | TeBDF | 2001- 2006 | (Granier et al., 2000) |
| 12 | FR-LBr | 44.72 | -0.77 | Cfb | CEF | 2003- 2006 | (Berbigier et al., 2001) |
| 13 | FR-Pue | 43.74 | 3.60 | Csa | TeBEF | 2001- 2006 | (Keenan et al., 2010) |
| 14 | IL-Yat | 31.34 | 35.05 | BSh | CEF | 2001- 2002 | (Grünzweig et al., 2003) |
| 15 | IT-Cpz | 41.71 | 12.38 | Csa | TeBEF | 2001- 2006 | (Tirone et al., 2003) |
| 16 | IT-Lav | 45.96 | 11.28 | Cfb | CEF | 2006- 2006 | (Marcolla et al., 2003) |
| 17 | IT-MBo | 46.02 | 11.05 | Cfb | TeH | 2003- 2006 | (Wohlfahrt et al., 2008a) |
| 18 | IT-PT1 | 45.20 | 9.06 | Cfa | TeBDF | 2003- 2004 | (Migliavacca et al., 2009) |
| 19 | IT-Ro1 | 42.41 | 11.93 | Csa | TeBDF | 2002- 2006 | (Rey et al., 2002) |
| 20 | IT-Ro2 | 42.39 | 11.92 | Csa | TeBDF | 2002- 2006 | (Tedeschi et al., 2006) |
| 21 | IT-SRo | 43.73 | 10.28 | Csa | CEF | 2003- 2006 | (Chiesi et al., 2005) |
| 22 | NL-Loo | 52.17 | 5.74 | Cfb | CEF | 1997- 2006 | (Dolman et al., 2002) |
| 23 | PT-Esp | 38.64 | -8.60 | Csa | TeBEF | 2002- 2006 | (Pereira et al., 2007) |
| 24 | PT-Mi1 | 38.54 | -8.00 | Csa | TeS | 2003- 2005 | (Pereira et al., 2007) |
| 25 | SE-Fla | 64.11 | 19.46 | Dfc | CEF | 2000- 2002 | (Lindroth et al., 2008) |
| 26 | SE-Nor | 60.09 | 17.48 | Dfb | CEF | 1996- 1997 | (Lagergren et al., 2008) |

[a] Koeppen-Geiger climate zone (BSh = hot arid steppe; Cfa = humid, warm temperate, hot summer; Cfb = humid, warm temperate, warm summer; Csa = summer dry, warm temperate, hot summer; Dfb = Cold, humid, warm summer; Dfc = Cold, humid, cold summer).

[b] Plant functional type (TeBEF = Temperate broadleaf evergreen forest, TeBDF = Temperate broadleaf deciduous forest, CEF = Coniferous evergreen forest, TeS = Temperate open woodland with C3 grass, TeH = C3 grassland).



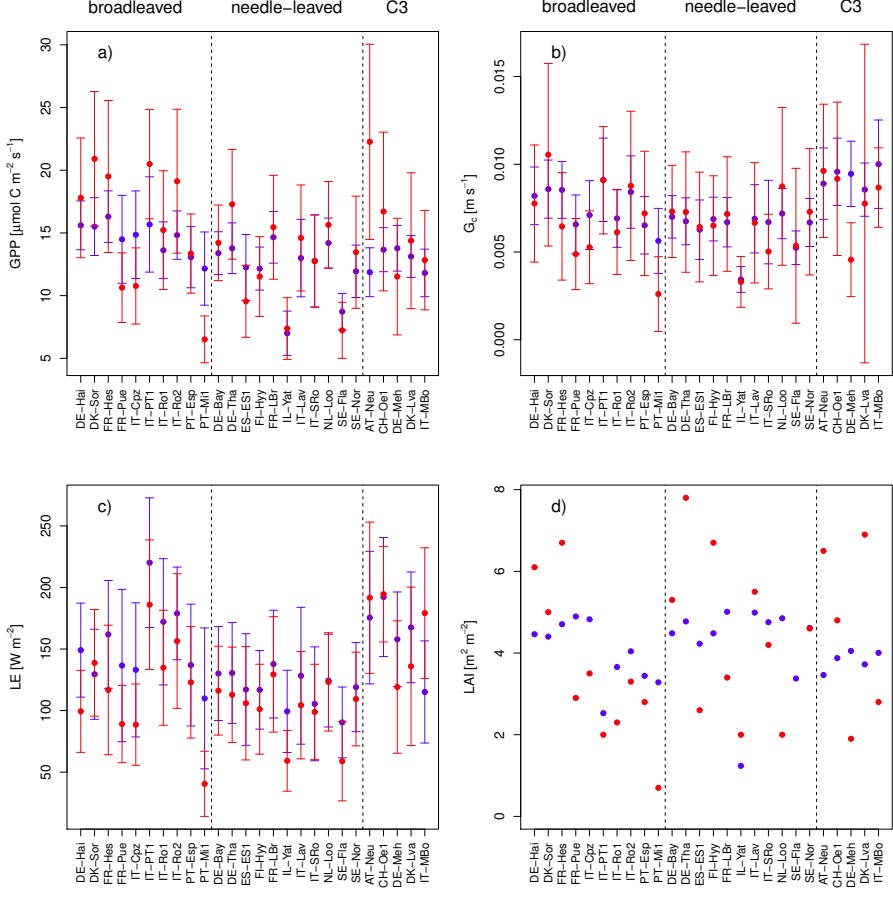

**Figure 10.** Comparison of measured a) GPP, b) $G_c$, c) latent heat fluxes (LE) and d) LAI at 26 European FLUXNET sites (red) and simulations by OCN (blue). Displayed are means and standard deviation of daily means of the measuring/simulation period, with the exceptions of FLUXNET derived LAI, which is based on point measurements.



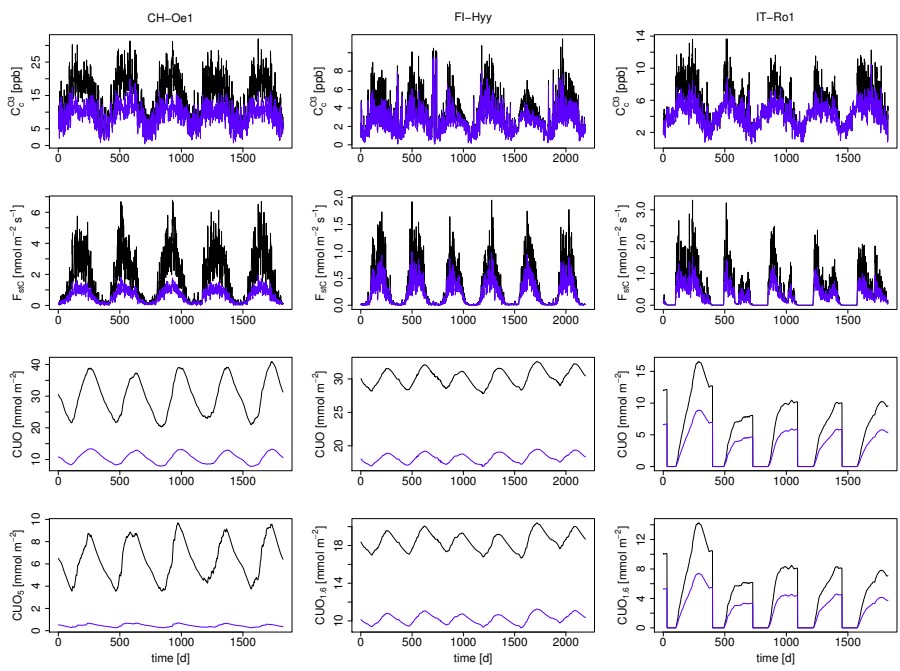

**Figure 11.** Plotted are the differences between the D, D-STO and ATM model version. Mean daily values of the a) ozone surface concentration [ppb], b) canopy integrated ozone uptake into the leafs [nmol m$^{-2}$ s$^{-1}$], c) canopy integrated cumulative uptake of ozone (CUO) [mmol m$^{-2}$] and d) canopy integrated cumulative uptake of ozone above a threshold (CUO$_5^{1.6}$) [mmol m$^{-2}$] for the three FLUXNET sites CH-Oe1, FI-Hyy and IT-Ro1. Blue: Difference between the D-STO model and the standard model (D), Black: Difference between the ATM model and the standard model (D).




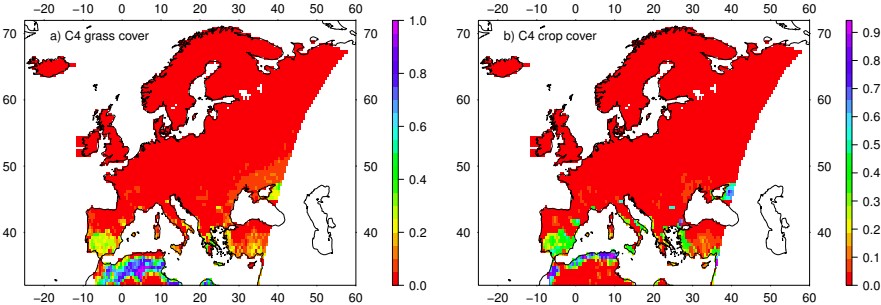

**Figure 12.** Europe-wide simulated mean cover fractions of C4 plant functional types for the years 2001-2010. a) mean simulated grid cell cover of the C4-grass PFT in OCN, b) mean simulated grid cell cover of the C4-crop PFT, both given as fractions of the total grid cell area.

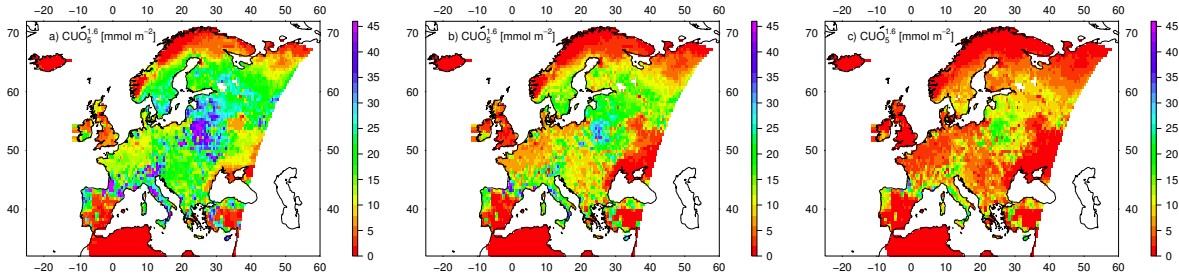

**Figure 13.** Mean decadal canopy integrated cumulative uptake of ozone above a threshold ($CUO_5^{1.6}$) $[\mathrm{mmol\,m^{-2}}]$ for Europe of the years 2001-2010. a) no ozone deposition scheme (ATM), b) ozone surface resistance is only determined by stomatal resistance (D-STO) and c) standard ozone deposition scheme (D).



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
