# Peer review of "Development and evaluation of an ozone deposition scheme for coupling to a terrestrial biosphere model"

_Biogeosciences, 2016_

## Referee Comment (RC1) · F. Dentener (Referee) · 17 Aug 2016

Review:

This is an interesting and well-written paper, describing assumptions and impacts of these assumptions regarding the inclusion of an ozone deposition scheme in terrestrial biosphere model on a number of ecosystem functions.

The authors make the point that the calculation of in-canopy (leaf-level) O3 concentrations is important for calculating stomatal uptake and impacts on photosynthesis- and this difference is one of the major causes of lower O3 impacts than suggested in other studies. A substantial part of the paper is about comparison of fluxnet data with

a number of underlying parameters, which are important to calculate ozone fluxes. I think this is a very good approach.

With some exceptions, the description of methods and results is quite clear, although my feeling is that the uncertainly analysis is covering only part of the overall uncertainties, and perhaps not all relevant processes are included. I recommend publication of this paper, when taking into account the remarks and suggestions below.

General remarks:

One of the key-equations (derived from Wittig et al., 2007) is equation 16. There are number of issues with the use of this regression equation.

1) As the authors remark in their discussion, a conceptual problem of using equation is that even at cumulative O3 uptake of zero, the equation still predicts a -6 % impact on photosynthesis. Also the slope of the equation -0.22 % per mmol m-2 is low compared to some other studies. I would suggest that refitting of the data, and forcing the values to go through zero is one option for a sensitivity study. Possibly another option is to re-fit these data to cumulative uptake above the threshold. On page 3/l. 28 the parameterization of Lombardozzi (2015) is mentioned, however without discussion on why this relationship is not used.

2) I would appreciate some discussion with regard of the validity of the experimental relationship for leaf-level ozone, or whether it also suffers from some atmospheric diffusion effect? Ideally when using such parameterizations the experimental conditions should be reproduced. I propose the authors have a look at some of the data used in Wittig, to resolve this issue.

3) I have difficulties to understand equation (15) page 8. Several issues need clariciation:

a) Why is Fst, detox used? The cumulative ozone uptake is dependent on the overall flux, regardless whether it is detoxified or not.

b) what is the rationale of using the factor fshed? Why would young leaves be less or not sensitive to ozone damage? What is the reference for this?

c) Rearranging this equation 15 gives CUO=c*Fst,detox*delta_t/fshed- I guess in times that fshed is close to zero, the values of CUO can get very large. I suspect something is not correct with this equation.

d) I would expect that CUO is something integrated over the canopy, as mentioned in p. 8 l 18- but it would be good to have the equation already describing this.

e) see remark 2) but the equation 16 seems to be valid for cumulated ozone flux, not for fluxes corrected for detoxication, as suggested by equation 15.

f) Somewhat related to the point above: even if plants can detoxify ozone, some costs will be associated with this mechanism. Where is the impact of this process accounted for?

4) Missing processes: there are several publications suggesting that ozone damage advances senescence (e.g. Gielen, 2007). Further ozone can damage of stomata-leading to slugginesh (e.g. Paoletti) To what extent are these processes included and how would they affect results?

5) Coupled atmospheric dynamics-vegetation ozone models suggest rather strong atmospheric responses and feedbacks. E.g. Super, Vilà‐Guerau de Arellano, Krol, JGR, 2015 as well as some papers cited here. I think the virtue of this publication is an increased understanding of the vegetation dynamic response (still with a lot of uncertainties), but in addition coupled atmosphere-vegetation simulations are still in its infancy. This should be clearly mentioned in abstract and conclusions.

Minor remarks:

p. 1 l.6 free troposphere is the region above the boundary layer. I guess the authors mean near-surface ozone in the planetary boundary layer.

p.1 l.9 Although it probably doesn't matter: are the authors comparing the model with or without ozone effects.

p. 1 l. 17- outside the leaves: suggest to call this near-leaves, or leaf-level ozone.

p. 2 l. 3 As raised in general comments, are the effects of anti-oxidant mechanisms somehow included?

p. 2 l. 4 Better to include a range: a factor 2 to 5. I personally do not think a factor of 5 is realistic.

p. 2 l. 11 delete 'less polluted' transport is taking place regardless of pollution levels.

p. 3 l. 18 'no damage' is observed. Detoxification: explain what consequences for productivity this can have.

p. 3 l. 28. Explain why this parameterization was not used

p. 4 l. 6 sensitivity analysis towards selected critical parameters?

p. 4. L. 9 accumulation of what?

p.4 l. 11-25 I would appreciate some more information on the models. How many canopy layers are in OCN? Is there an interaction of N in leaves with ozone? What version of the EMEP model (output) was used, regional, global, resolution? Explain vertical structure of EMEP- can a constant mid-of-the gridbox of 45 meter be safely used, or are the regions (e.g. in the mountains) where this value is different (i.e. is the coordinate system fixed altitude, pressure or hybrid)? I think the model can also output near-surface ozone (diagnostic). Why was this not used- it would avoid additional uncertainty in the recalculation of the atmospheric resistance.

p. 5 Ra is the resistance between the surface (near-canopy) and 45 meter (i.e. it is not at a level of 45m).

p. 4 l. 18 Can something be said on how this conductance is distributed over the

canopy layers- in general how vertical canopy structure is expected to influence ozone uptake.

p. l. 19: was this calibration necessary for this study, or more general for OCN model results?

p. 6 mention which three PFTs were considered for this LAI+1 approach? Probably for the readers not wanting to go back to older papers, a table listing some characteristics of the PFTs (appendix) would be useful.

p. 6/7 eq. 8,9,10 to what extent are these equations based on observations, or merely model assumptions (and what is the associated uncertainty).

p. 7 l.18 1 mmol/m2: is this referring to cumulative ozone uptake? Is this published (reference?). I am not sure if such sensitivity with an atmospheric model which would include chemistry feedbacks, can be translated into such small uncertainty for the vegetation o3 uptake. Note that there is in general a quite large difference in PBL mixing in a variety of atmospheric models- which in itself already suggests a large uncertainty.

p. 8 l-1-20 See remarks above- need to get a better description if/how detoxicifaction is included.

p. 8 l. 8: do I understand correct that in the rest of the text CUO1.6/5 would refer to equation 15; while CUO would refer to use Fst in equation 15. This needs to become clear- and the correct equations need to be given.

p. 9 l. 5. Would a sensible variation of dl (equation 16) also be a critical parameter? How was this subset of parameters selected.

p. 9 l. 22 What is the La Thuille dataset?

p. 9 l. 25 How many years were need to reach equilibrium? What was the criterium for equilibrium?

p. 9 l 28: Which EMEP simulations were providing this 100 years transient concentrations? Is there a reference?

p. 9 l. 29 Appendix tab 1? I think just table 1.

p. 10 I understand that the purpose of section 2.4 is to derive trust in the model, when testing to observable parameters. I would however need some more insight in why morning/evening fluxes need to removed, and data with the different soil moisture. What would be the effect of not removing such data?

p. 10 l. 29- brought into equilibrium. How done?

p. 11 figure 1: obviously the largest discrepancy is found for LAI and in p. 12/l. 18 the authors suggest that this is not important. How is it possible to have realistic GPP etc; and such a spread in LAI? Please explain.

p. 12 l. 24. While it is facilitating the discussion to focus on only 3 stations, some words on how representative these stations were for others would be welcome.

p. 13 l. 5-25 I would advise to also see Hardacre, Atmos. Chem. Phys., 15, 6419-6436, 2015, for further opportunities to evaluate the ozone deposition velocities and fluxes.

p. 13 l. 35 Can you confirm that sapflow measurements are not reliable for this study?

p. 14 repeat here that Fr is the ratio of stomatal to overall flux. It would be interesting to give average values (per ecosystem/PFT) over the months. Perhaps for an appendix? I think this could be useful for comparison in future studies.

p. 14 l. 34 I didn't quite understand the sentence . . .not zero . . .because accumulate over several years. Isn't it simply that there is already some photosynthesis activity?

p. 15 l. 8-10. It is not clear to me whether OCN has croplands, and if so what crop? The authors mention C3 crops- I guess that would be mainly wheat?

p. 15 l. 17 Figure 6a is an EMEP model output?

p. 15 Appendix 12 ab missing. Do you mean Figure 12? See=>sea.

p. 15 l. 34 interesting dynamical/phonological feedback, but it also reminds that things like early senescence are probably not included.

p. 16 section 3.5 Please remind reader of what D-STO and ATM were? See section 2.6. Appendix 11 and 13 are missing. L. 13 uptake and accumulated: rephrase in: accumulated uptake

p. 16 spell out the meaning (remind the reader) of CUO1.5 and 5.

p. 17 section 4.1 l. 10. Interactions with VOCs (as well as soil NOx emissions, see Ganzeveld's paper), are important. But I don't understand how they are implicitly included, especially in the OCN framework.

p. 17 l. 20: was O3 needed to reach this good agreement. Probably not- explain.

p. 19 l 3. As explained above, I think this warrants some additional analysis.

p. 19 l. 22 impacted=>determined?

Figures:

Figure 6: Why are the units of panels b and c different? The chosen range doesn't work well for panel b (all purple).

Figure 7: it is hard to discriminate the colors in Figure 7.

Figure 9: legenda describing a) can be improved.

Figure 12: color scheme doesn't work (mostly red)- more resolution for low values is need (0-10 %). For C4 crops- is irrigation consired?

---

## Referee Comment (RC2) · Anonymous Referee #2 · 1 Oct 2016

The authors present the development and application of a new scheme for estimating stomatal and non-stomatal deposition of ozone within a coupled carbon-nitrogen cycle model. the performance of the model is evaluated against gross primary productivity and latent heat fluxes measurements and estimates of canopy conductances at 26 FLUXNET sites covering a broad range of geographical regions and ecosystem types. The model is found to perform well and simulations are performed to assess cumulative ozone uptake and the resulting reduction in GPP and transpiration at 3 sites (one broadleaf, one needleleaf and one C3 grassland site). They are able to show a clear difference between estimates of ozone uptake and damage made using this scheme and those in which damage is assessed using concentration-based analyses. Overall

this is a well-planned and well-executed study that is both timely and highly topical, addressing an issue of real concern to society. I strongly recommend publication in Biogeosciences once the following comments have been addressed:

Overall, I find that the manuscript is too detailed and too verbose, with the authors often repeating a point several times, and would benefit from some substantial restructuring. In particular I would recommend that the authors combine Sections 3 and 4 into a single "Results and discussion" section as much of the discussion in Section 4 is merely a repetition of issues raised in Section 3 and in many places reads more like a conclusion. In this case, the conclusion could be made longer. In addition, many of the results presented and discussed in Section 3 are a distraction to the main message (the improvement in estimates of ozone damage when using this new deposition scheme) and would be better moved to Supplementary Information. Specific comments and suggestions follow.

The authors use "ozone" and "O3" fairly randomly throughout the manuscript. I would suggest sticking with one or the other.

Abstract P1, L6 - This is the first use of the acronym OCN - please explain what it is. P1, L12 - "update" should read "uptake" P1, L15-6 - Please re-word, this is hard to follow. I think that you are saying: "When applied at the European scale, we find that including our new ozone deposition scheme substantially affects simulated ozone…."

Introduction P2, L22 - replace "consequence" with "result" P2, L24 - replace "extend" with "extent" P2, L27-29 - I suggest making the point here that AOT40 is currently used for regulatory assessment purposes in Europe. P2, L32-33 - Please could the authors explain what they mean by "regional provenances". Do they mean that the same species in different geographical locations differ? Or that different regions have different ecosystems? P3, L8 - Up until this point the authors have referred to AOTX. As AOT40 is the regulatory metric and one that they use in subsequent analysis and discussion I suggest they clearly define AOT40 at this point. P3, L23 - I suggest the

authors make the point that the threshold values are species-specfic to account for plant sensitivity/tolerance to ozone.

Methods P4, L20 - The model acronym EMEP MSC-W should be defined here rather than at the end of the paragraph, e.g. "The ozone and N-deposition data used for this study are provided by the EMEP MSC-W (European Monitoring and Evaluation Programme Meteorological Synthesising Centre - West) chemical transport model (CTM; Simpson et al., 2012a)." P4, L22 - insert "been" between "have" and "documented" P5, L1 - replace "in" with "at" and remove "height" P5, L7 - replace "in" with "at" and remove "height" P5, L15-6 - replace "leafs internal" with "internal leaf" P5, L16 - parentheses should only be around "2005" P5, L17 - replace "ozone to water vapour" with "ozone from water vapour" P5, L19 - is this factor of 0.7 included in Zaehle and Friend or is this new for this current study? P6, L11 - please explain more clearly what is meant by a low temperature correction factor and why it is needed. P6, L11 - suggest rewording to: "is scaled by a low temperature correction factor, FT, such that" P6, L13 - suggest rewording to: "where TS is the 2m air temperature (C; Simpson et al., 2012a, eq. 60) and 1<FT<2." P6, L20 - replace "Like" with "As" P7, L1 - parentheses should only be around "2003" P7, L4 - suggest combining to give: "0.5, to prevent negative values in the first fraction of eq. 10". P8, L4 - Why PODI? My understanding of PODY is that the Y stands for the threshold value not the canopy level. P8, L13 - What is the physical (real-world) interpretation of the parameters 0.22 and 6.16 in eq. 16? P8, L13-4 - Why not just divide by 100 in the equation itself? P8, L17 - Please explain to the general audience why a reduction in An results in reductions in Gst and (particularly) Ci. It is not intuitive why this would reduce internal concentrations. P8, L23 - parentheses should only be around "2010" P9, L2-3 and throughout - I would suggest that the authors re-define or at least use a word description each time these parameters are re-introduced at the start of a new section; else provide a table listing the key parameters for the reader to refer back to. P9, L11 - Are the "summer months" defined here the same as what is then referred to as the "growing season"; if so, please make clear, if not, please define growing season separately. P9, L21 - Please explain what is

meant by "site levels". Is this "site-specific" i.e. OCN is run as a column model rather than a 3-D regional model? P9, L22 - square parentheses are not required around $CO_2$ as the text includes the word "concentrations". P9, L23 - parentheses should only be around "2015". P9, L23 - rearrange this to read: "Reduced and oxidised nitrogen deposition in wet and dry forms and hourly ..." P9, L27 - O3 should be subscript P9, L28-9 - Why not use GCM output or reanalyses data where there is a lack of observation data? P9, L30 - what do the authors mean by time-varying here? Surely the progressive simulations also used data that varied with time. Do the authors mean that here it is observations from the site in question for the years in question? P10, L2 - Why have the authors chosen to base LAI on single point, time-specific observations rather than e.g. MODIS LAI data? It seems that this introduces a considerable source of uncertainty. P10, L5 - parentheses should only be around "2015" P10, L6-7 - suggest rewording to read: "...are filtered prior to deriving average growing-season fluxes to reduce the effect of model biases on the model-data comparison. Night-time and ..." P10, L9 - please explain what a "modelled soil moisture constraint factor" is, and why a threshold of 0.8 has been chosen as a filter. Is this based on observations suggesting severe drought impacts alter fundamental plant functioning? P10, L10-1 - suggest rewording to "Daily mean values are calculated from the remaining time steps only where both modelled..." P10, L14 - why only use July here when the rest of the analysis is conducted for JJA? P10, L14-15 - why not use the same light level to define daylight as you used to filter the data previously? P10, L16 - suggest rewording to "..FR and for both modelled and FLUXNET-observed GPP..." P10, L22-3 - suggest rewording to "...1999). Reduced and oxidised nitrogen deposition in wet and dry forms and ozone..." P10, L25 - parentheses should only be around "2014b" P10, L25 - insert "and are" before"scaled back" P10, L27 - parentheses should only be around "2011" P10, L28 - square parentheses are not needed around $CO_2$. P10, L28 - parentheses should only be around "2015" P10, L29-30 - Please check dates. If 1961-1970 is used as a spin-up shouldn't the simulation then start at either 1961 (repeating the first 10 years) or from 1971? P10, L32 - Please explain what an MTE product is. P11, L2 -

replace "Different" by "In contrast" P11, L3 - O3 should be subscript P11, L3-4 - Please explain for the non-specialist audience why the resistances result in a lower canopy concentration.

Results P11, L11 - what do the authors mean that they agree "within the standard deviations"? Are they stating that the data overlap? It would be better to demonstrate this goodness of fit with robust statistical analysis. P11, L13 - should read "...very close, with only slight under-" P12, L3 - remove extra ")" after 10 a P12, L5-6 - please give an example of site management that might result in such variability P12, L8 - why should LE be overestimated and GPP underestimated by OCN at broadleaved forest sites? P12, L13 - what do the authors mean by "vary more widely"? Do they mean that there is a greater difference between modelled and measured values or that there is greater variability in the differences? P12, L14 - Do the means still lie within one standard deviation or not? Is there a tendency for the model to consistently under- or over-estimate? P12, L15-22 - move to SI P12, L23 - general comment regarding section 3.2: Do the reported "biases" in the diurnal cycles reflect those of the means? i.e. is GPP underestimated at the broadleaf site. P12, L24 - diurnal profiles of which variables? State here P12, L32 - remove unnecessary parentheses after m and n. P12, L32 - should read: "with particularly good agreement..." P12, L32 - surely it's more relevant that it is an evergreen needle-leaf forest that it is Finnish? P12, L34 - again, state the type of landcover at this site P13, L1 - Again please explain what is meant by the means being within the standard deviation. P13, L2 - The maximum variability at CH-Oe1 seems to occur during the middle of the day P13, L3 - "whereas" is all one word P13, L4 - what about the peak GC at the CH-Oe1 site? Is it also overestimated by the model? P13, L5 - "simulate" rather than "simulated" P13, L5-6 - is this not a serious short-coming of the model water response parameterisation? I thought the midday depression in GC was a well observed response to water stress. Please comment on the likely implications for your results and conclusions? P13, L7-> Please either change the order of the panels in Figure 2 or the order of the text so that you are presenting the results of the panels in the order in which they appear. P13, L9-

15 - How is RC measured? or is it back-calculated from observed ET and LE? Please comment on the reliability of the observations. P13, L9-15 - what are the implications of the model deviations from observations? P13, L15 - should read "...observed which is slightly lower..." P13, L16 - the minimum velocities appear to be lower than this value for crops P13, L18 - "barely" should read "barley" P13, L16-20 - The modelled velocities at your crop site are well below these. P13, L20 - please rephrase to "The estimates for Hyytiälä also agree..." P13, L16-23 - It would be helpful if you compared the data site by site as before P13, L23 - Why is Vg so noisy for IT-Ro1? P13, L24 - Perhaps it is worth making the point that Vg is not zero because of non-stomatal deposition. P13, L27-28 - Why is there such large variability in the afternoon at IT-Ro1? Is that another sign of water stress? P12-13 - general comments: For Rc, Vg, FR, FStC: what are typical/expected profiles of these variables? Do we really only have observations at 1 or 2 times per day with which to assess model skill? How do these output data compare with estimates from other models? I would strongly recommend that much of the content here is moved to SI and/or presented in a table, with this section only highlighting a few key or interesting features. P14, L2 - add a reminder in the parentheses that GCO3=GC/1.51 P14, L3 - Is this ratio essentially the proportion of deposition that is stomatal? P14, L3-9 - Why have the authors chosen to report the 24-hour average for this variable and not for the others? Section 3.3 This section and the accompanying figure should be moved to SI, with only a few key headline findings included in the main text. P14, L12 - replace "constraint" with "constrained" P14, L13 - "boreal" would be a more useful descriptor than "Finnish" P14, L13 - replace "except of" with "except for" P14, L14 - replace "describing" with "which describes" P14, L17 - replace "compared" with "relative" P14, L22 - insert "canopy conductance" before "GC" P14, L23 - replace "what causes" with "resulting in" P14, L24 - replace "compared" with "relative" P14, L25 - remove "changed values for" P14, L26 - explain the units (%/%) P14, L27 - remove "very" and "varying" P15, L1-2 - has this phenomena (the effect of needle-shedding on CUO) been evaluated? P15, L6-7- what percentage is 250 gC/m2/yr? P15, L8 - remove "to this acceptable agreement" P15, L9 Again what

percentage is 400 to 900 gC/m2/yr? P15, L12-3 - It also makes it difficult to assess the reliability of the model! P15, L16 - Please explain how N limitation can lead to overestimation of GPP P15, L20 - Fig. 6d does not show GPP. Should this read Fig. 5a? P15, L23-4 - Is it not to be expected that AOT40 closely follows absolute ozone concentrations? P15, L26 - replace "averaged" with "ranged from 60 to 120" P15, L27 - move "(Fig 7 a)" to between "Europe" and "and" P15, L28 - "larger" should read "large" P15, L28 - does this refer to Fig. 7b? P15, L29 - suggest rewording: "with high cover of C4 PFTs, e.g. Black Sea area (see Appendix 12 a,b)." P15, L30-1 - suggest rewording: "...where productivity is low and stomatal O3 uptake reduced by low O3 concentrations or drought control on stomatal fluxes respectively." P15, L31-2 - suggest removing the sentence beginning: "Slight increases or strong decreases..." P15, L32 - "increases" should read "increase" P16, L3 - replace "by" with "of" P16, L4 - insert "Fig. " before "7 c" P16, L4 - insert "of transpiration" after "3-4%" P16, L4 - remove "to" before "4-6%" P16, L5 - insert "relative" before "reductions" P16, L7 - should read "Black Sea" P16, L8 - insert "Fig." before "7 d" and replace "They are" with "These are" P16, L10 - please explain why a reduction in transpiration matters. P16, L15 - suggest rewording: "...CUO1.6 increases more strongly by 35%..." P16, L18-9 - It seems to me that in this study simulation D is effectively the base case and D-STO and ATM are sensitivity tests. It would therefore make more sense to swap panels a and c in Figure 9. Furthermore, it seems to me that this is the real headline message of this study - that the ozone deposition scheme substantially alters estimates of impacts. this needs far more emphasis (it is currently hidden by the wealth of detail in the rest of this discussion) and Figure 9 should include further panels showing how CUO changes (see below).

Discussion This section seems redundant. Much of it is either already stated in the Results section or could be moved to form part of a more robust conclusion. P16, L24-5 - replace "with the aim" with "in order to" P16, L25 - replace "effect to net" to "effect on net" P16, L25 - remove "the" before "regional" P16, L28 - replace "assuming" with "the assumption" P16, L28 - replace "would be identical" with "is identical" P16, L29

- replace "in 45m" with "at 45m" P16, L30-1 - suggest rewording: "...and deposition variables i.e. calculated ozone uptake ..." P16, L32 - P17, L2 - suggest rewriting: "Our sensitivity analysis does show that a correct estimate of canopy conductance is crucial for calculating plant ozone uptake. We find that the model produces reasonable estimates ..." P17, L2 - replace "a range of" with "some" P17, L7-8 - suggest rewriting: "Reliable estimates of surface ozone concentrations are also essential for calculating canopy ozone uptake FstC" P17, L8-9 - suggest rewriting: "...airspace due to biogenic volatile organic compounds (BVOCS) emitted by vegetation is (at least partly) implicitly included in the" P17, L9-10 - Does this mean there is a degree of double accounting? P17, L11 - suggest "performance" or "efficacy" in place of "functionality" P17, L15 - suggest combining these to form a single sentence: "...changes in GC emphasising the importance..." P17, L15-16 - How can reliable estimates be obtained? P17, L18 - replace "indicates"with "indicate" P17, L26 - replace "impose" with "introduce" P17, L29 - replace "suitable" with "well able" P17, L30 - remove first occurrence of "finding" and replace "encourages" with "supports" P18, L2 - reword: "Estimates of the regional damage to annual average..." P18, L2 - make clear this is transpiration rather than temperature (I assume) P18, L2-3 - remove "the period of the years" P18, L3 - replace "lower" with "low" and "previous" with "previously" P18, L3 - should read "Meta-analyses" and "an 11%" P18, L6 - should read "Land Model" P18, L7 - reword: "..transpiration have been estimated as 5-20% for Europe and 2.2% globally..." P18, L9 - reword: "plant types. Damage was only related to cumulative ozone uptake for one plant type with a very small slope" P18, L9 - please explain the real-world meaning of a small slope. P18, L14 - use "discrepancies" or "differences" rather than "deviations" P18, L14-15 - replace "the usage of very different" with "differences in" and then remove "different", "differing" and "non-identical" P18, L16 - replace "differences in simulating" with "simulation of" P18, L17 - reword: "The key difference from the previous study is our use of the ozone..." P18, L17 - remove "included in our study" P18, L21 - remove "the" before "non-stomatal" P18, L22 - should read "To obtain as accurate as possible an estimate..." P18, L23 - replace "it's" with "their" P18, L24 - replace "considered" with

"accounted for" P18, L25 - suggest moving "(possibly PFT specific)" to come before "flux threshold" P18, L25 - "it's" should read "its" p18, L25 - should the "Y" in "CUOY" be a subscript? P18, L32 - insert "see" before "LRTAP" P18, L33 - replace "but only" with "there are" and "exists for" with "of" P19, L2-4 - What is the implication of this disadvantage to the findings reported here? P19, L5 - replace "damage estimates" with "relationships" P19, L6 - replace "estimates" with "metrics" P19, L13 - replace "should be regarded too" with "also requires further analysis"

Conclusion This section needs to be substantially expanded. The authors would also do well to identify (even using bullet points if necessary) the key findings of their study and the implications for the land surface and atmosphere research communities. Much of Section 4 could be distilled and included in the Conclusion section.

P19, L20-1 - replace "to generally consider" with "that" P19, L21 - reword: "non-stomatal ozone uptake is routinely included in model assessments of ozone damage . . ." and remove "estimate" after "better" P19, L22 - remove "used" P19, L23 - insert "used here" after "scheme" P19, L23 - reword: "importance of reliable modelling of canopy conductances as well as realistic. . ." P19, L24 - insert "as" before "accurate" P19, L26 - remove "Desirable are" P19, L27 - insert "are also desirable" after "types" P19, L29 - replace "regarded" with "considered" P19, L29 - insert "," after "thresholds"

Appendix A P20, L1 - capitalise "Aerodynamic Resistance" and remove "(Appendix material)" P20, L3 - remove "," after "heights" and replace "This data is" with "These data are" P20, L4 - replace "in 45m height" with "at 45m" P20, L7 - what does U10 mean? If at 10m, why is this an appropriate height at which to calculate u*? P20, L9 - replace "in 45m height" with "at 45m"

Appendix B P20, L21 - Why not use ORCHIDEE to calculate biogenic emissions? P20, L22 - remove "NO from" P20, L24 - Volcanic emissions of what? Which compounds?

References Please check references carefully. Tuovinen et al., 2004a and 2004b are the same paper Tuovinen et al., 2009a and 2009b are the same paper

Figures: Throughout - I would suggest that rainbow scale is not the most effective and that limited colour graduated scales would be easier to interpret.

Fig. 1 Panel (d) - Again, why choose a non-varying measure of LAI (i.e. point samples) rather than MODIS or similar, particularly as you comment on the validity of these measurements for the specific time period modelled? Panel (d) - In its present form this is not a useful panel and I would suggest that it is removed or moved to SI. It distracts from the good fit the model shows to other (more important) variables. Caption - line 4 should read "...which are based on point..."

Fig. 2 x-axis scale - Hours should have a 4-hour or 6-hour scale, not 5. Please state explicitly whether this is local time or UTC. y-axis scale - As the scale is the same across each row I would suggest only one axis scale is required. y-axis scale - for variables that can be negative please add a dashed horizontal line to indicate 0.0; otherwise the axes should cross at zero.

Fig. 3 scales - please define the scales used in Fig 3 more carefully, either here in the caption or in the appropriate place in the main text.

Fig. 4 This figure should be SI. In addition, it is virtually unreadable. I had to view at 600% zoom to make out the yellow and red lines

Fig. 5 scales - don't use the same colour scales for both absolute values and changes; changes are best shown on blue-red scales. Use e.g. green scale for crop cover.

Fig. 7 scale - please improve the scales; I suggest using a graduated single or limited colour range. panel labels - please use more descriptive panel captions (not just "damage")

Fig. 9 To me, this is the KEY figure in this paper. I suggest that you add panels showing changes in CUO from D to D-STO and ATM respectively (giving a 5 panel plot)

---

## Author Comment (AC1) · 9 Nov 2016

Dear Referee,

we thank you for your detailed and constructive comments that helped considerably to improve the manuscript.

Yours Sincerely
Martina Franz

**1 General remarks**

One of the key-equations (derived from Wittig et al., 2007) is equation 16. There are number of issues with the use of this regression equation.

Q: 1) As the authors remark in their discussion, a conceptual problem of using equation is that even at cumulative $O_3$ uptake of zero, the equation still predicts a -6 % impact on photosynthesis. Also the slope of the equation -0.22 % per mmol m-2 is low compared to some other studies. I would suggest that refitting of the data, and forcing the values to go through zero is one option for a sensitivity study. Possibly another option is to re-fit these data to cumulative uptake above the threshold. On page 3/l. 28 the parameterization of Lombardozzi (2015) is mentioned, however without discussion on why this relationship is not used.
A: A refitting of the Wittig damage function would be desirable. However, a data request to V. Wittig remained unanswered. A refitting can not be done without repeating the work done by the meta-analysis.
There are several reasons for not using the Lombardozzi damage function.
For tree species, Lombardozzi et al. (2015) assume a fixed reduction of net photosynthesis due to ozone independent of the actual ozone uptake. This fixed reduction is -12.5 % for broadleaved species and -16,1 % for needle-leaved species. Only for crops and grasses ozone damage to net photosynthesis depends on ozone uptake. In other words, the atmospheric ozone concentration and ozone uptake into the plants do not affect the damage estimate for tree species but only for grasses and crops. Due to the lack of impact of ozone uptake on ozone damage estimates, the offset implied by Lombardozzi et al. (2015) is actually higher. The effect of the step decrease in Lombardozzi et al. (2015) might be ameliorated by the fact that canopy conductance is affected in parallel. However, this results in a general decoupling of photosynthesis and canopy conductance. Our aim here was to investigate the effect of

ozone damage to net photosynthesis under the assumption that photosynthesis and canopy conductance remain coupled. We have extended the discussion to make this point clearer.

Q: 2) I would appreciate some discussion with regard of the validity of the experimental relationship for leaf-level ozone, or whether it also suffers from some atmospheric diffusion effect? Ideally when using such parameterisations the experimental conditions should be reproduced. I propose the authors have a look at some of the data used in Wittig, to resolve this issue

A: The experiments used by Wittig et al. (2007) do not use the leaf-level ozone concentration to calculate ozone uptake but the atmospheric ozone concentrations. The ozone uptake calculation thus differs in this respect between our simulations and the experiments used to derive the damage relationship. However in the experiment, ozone uptake is not directly measured. Rather, it is calculated from mean ozone concentrations over the exposure period and the respective average stomatal conductances. Thus the estimated ozone uptake rates and hence the amount of accumulated ozone used to derive a damage relationship are a coarse approximation and underlie considerable uncertainty. Following this the error introduced by using leaf-level ozone concentrations instead of atmospheric concentrations seems small, especially since the use of the leaf-level ozone concentrations is the physiological more appropriate approach. We have extended the discussion to make this point clearer.

Q: 3) I have difficulties to understand equation (15) page 8. Several issues need clarificiation: a) Why is Fst, detox used? The cumulative ozone uptake is dependent on the overall flux, regardless whether it is detoxified or not.

A: The Wittig damage function bases on CUO which accumulates the ozone uptake without a threshold. We changed this equation and rerun all simulations. In the new version ozone damage is calculated on ozone uptake accumulated without a

threshold. We note that this does not affect any of our conclusions, but agree with the reviewer that this is a cleaner way to address the issue.

Q: b) what is the rationale of using the factor fshed? Why would young leaves be less or not sensitive to ozone damage? What is the reference for this?
A: $f_{shed}$ is the fraction of new developed leaves per time step and layer. In the revised version, this factor was renamed to ($f_{new}$) to facilitate the understanding. New grown leaves are assumed to be undamaged. For evergreen species the old damaged leaves still exist when new leaves are grown. In this condition, $f_{new}$ causes the canopy layer CUO to be reduced when new leaves are grown, because they are health do not suffer ozone damage yet, i.e. if 10 % new leaves are grown ($f_{new} = 0.1$), the CUO is reduced by 10 %. Without this equation, newly grown leaves would be assumed to be similarly damaged to already existing foliar, which is not correct, and would cause the CUO for evergreen species would continuously increase over the years.

Q: c) Rearranging this equation 15 gives CUO=cFst,detoxdelta_t/fshed - I guess in times that fshed is close to zero, the values of CUO can get very large. I suspect something is not correct with this equation.
A: The equation was rewritten as:

$$\frac{dCUO_l}{dt} = (1 - f_{new})CUO_l + cF_{st,l} \tag{1}$$

As already mentioned in b). $f_{new}$ (formerly $f_{shed}$) s the fraction of new developed leaves per time step and layer. $f_{new}$ can take values between zero and one. $f_{new} = 0$ when no leaves are grown in the present time step, and $f_{new} = 1$ when newly grown leaves make up all of the present canopy. The $CUO_l$ of the previous time step is reduced according to the fraction of new grown leaves $(1 - f_{shed})CUO_l$.

Q: d) I would expect that CUO is something integrated over the canopy, as mentioned in p. 8 l 18- but it would be good to have the equation already describing this.
A: An additional equation clarifying this was added (new eq. 15):

$$CUO = \sum_{l=1}^{n} CUO_l. \tag{2}$$

Q: e) see remark 2) but the equation 16 seems to be valid for cumulated ozone flux, not for fluxes corrected for detoxification, as suggested by equation 15.
A: Yes. Equation 15 was adapted to use $F_{st}$ (without a threshold), and all simulations are rerun. Plots containing $CUO^1.6_5$ are substituted with CUO of skipped. Equation 14 was skipped.

Q: f) Somewhat related to the point above: even if plants can detoxify ozone, some costs will be associated with this mechanism. Where is the impact of this process accounted for
A: Costs for detoxification are not accounted for in the current model version. To our knowledge no suitable data are available to parametrise e.g. the increased respiration costs according to ozone uptake, since it is very hard to disentangle costs for ozone detoxification from other factors influencing leaf respiration under elevated ozone exposure.

Q: 4) Missing processes: there are several publications suggesting that ozone damage advances senescence (e.g. Gielen, 2007). Further ozone can damage of stomata- leading to sluggishness (e.g. Paoletti) To what extent are these processes included and how would they affect results?
A: Reduction of photosynthetic capacity is one feature of early senescence, others are not included. Omitting effects like early litter fall will underestimate ozone damage.

Stomatal sluggishness is not included in the model version described here. Transpiration rates are thus underestimated compared to accounting for sluggishness. A model version of OCN exists where sluggishness can be accounted for however in this case it occurs permanently for all PFTs. This seems to overestimate the effect at least in regions where low ozone concentrations occur. Following this stomatal sluggishness is an important aspect of ozone damage however it seems not reasonable to generally include it in the base model version. The simulation of sluggishness might be very interesting in a sensitivity study where also other effects like detoxification (e.g. through various flux thresholds) are tested on their impact on ozone damage estimates. We have extended the discussion to clarify that the current model does not include all known ozone effects.

5) Coupled atmospheric dynamics-vegetation ozone models suggest rather strong atmospheric responses and feedbacks. E.g. Super, Vilàâ RGuerau de Arellano, Krol, JGR, 2015 as well as some papers cited here. I think the virtue of this publication is an increased understanding of the vegetation dynamic response (still with a lot of uncertainties), but in addition coupled atmosphere-vegetation simulations are still in its infancy. This should be clearly mentioned in abstract and conclusions.
A: We add this point to discussion and conclusion. However, we don't think that this issue is important enough to merit mentioning in the abstract.

**2  Minor remarks**

Q: p. 1 l.6 free troposphere is the region above the boundary layer. I guess the authors mean near-surface ozone in the planetary boundary layer
A: Is changed to 45 m height.

Q: p.1 l.9 Although it probably doesn't matter: are the authors comparing the model with or without ozone effects
A: The model 'including $O_3$ damage' is used.

Q: p. 1 l. 17- outside the leaves: suggest to call this near-leaves, or leaf-level ozone.
A: Is called 'leaf-level ozone'.

Q: p. 2 l. 3 As raised in general comments, are the effects of anti-oxidant mechanisms somehow included?
A: No, since the flux threshold is omitted in the final version no detoxification occurs.

Q: p. 2 l. 4 Better to include a range: a factor 2 to 5. I personally do not think a factor of 5 is realistic.
A: Changed to 'a factor 2 to 5'.

Q: p. 2 l. 11 delete 'less polluted' transport is taking place regardless of pollution levels.
A: Done.

Q: p. 3 l. 18 'no damage' is observed. Detoxification: explain what consequences for productivity this can have
A: Detoxification causes increased respiration costs and following this reduces NPP what may reduce growth and biomass.

Q: p. 3 l. 28. Explain why this parameterization was not used

A: Atmospheric ozone concentrations and cumulated $O_3$ uptake only impact net photosynthesis of one plant functional type directly. For the two other plant types net photosynthesis is reduced in a step function independent of the accumulated ozone uptake.

Q: p. 4 l. 6 sensitivity analysis towards selected critical parameters?
A: The aim of the sensitivity study is to test the functionality of the deposition model, because it is calculates leaf-level $O_3$ concentrations and hence has a large impact on $O_3$ uptake estimates. The variable $R_b$ is also an important variable of the deposition model and was added to Fig. 3. The respective sentence on p 4 was changed from "provide a sensitivity analysis of the model to evaluate the reliability of simulated values of $O_3$ uptake' ' to 'provide a sensitivity analysis of critical variables and parameters of the deposition model to evaluate the reliability of simulated values of $O_3$ uptake'.

Q: p. 4. L. 9 accumulation of what?
A: Accumulation of ozone. Changed to '$O_3$ uptake and cumulated uptake'.

Q: p.4 l. 11-25 I would appreciate some more information on the models. How many canopy layers are in OCN?
A: There are maximum 20 layers. The number of actual simulated layers depends on the site and the PFT. Included in Methods section.

Q: Is there an interaction of N in leaves with ozone?
A: Yes. Photosynthetic capacity depends on leaf nitrogen concentration and leaf area, which are both affected by ecosystem available N. Increases in leaf nitrogen content enable higher net photosynthesis and higher stomatal conductance per unit leaf area. This in turn affects transpiration as well as ozone uptake and ozone damage

estimates. Included in Methods section.

Q: What version of the EMEP model (output) was used, regional, global, resolution? Explain vertical structure of EMEP- can a constant mid-of-the gridbox of 45 meter be safely used, or are the regions (e.g. in the mountains) where this value is different (i.e. is the coordinate system fixed altitude, pressure or hybrid)?

A: We used version 4.4 of the EMEP MSC-W model, in essentially the same setup described in the nitrogen deposition study of Simpson et al. (2014b). The model was run for the regional RCA3 domain, driven by RCA3 meteorology. The vertical structure is the standard EMEP one (see Simpson et al., 2012) with a lower layer of about 90m thickness. The coordinates are terrain following (sigma coordinates) though, so the mid-point of ca. 45m is relative to the assumed ground surface in such a system. The main assumption of all EMEP deposition modelling is that this 45m height lies within the surface layer, so that standard similarity theory can be applied. This assumption is not always correct of course, but in general the EMEP model's predictions of near-surface ozone (and even fluxes, e.g. Klingberg et al, 2008) suggest that the methodology is reasonable.

Q: I think the model can also output near-surface ozone (diagnostic). Why was this not used- it would avoid additional uncertainty in the recalculation of the atmospheric resistance.

A: The calibration of leaf-level ozone concentrations is impacted by e.g. LAI and canopy conductance. They both determine the amount of ozone taken up by the stomates and hence reduce the ozone concentration within the canopy air. EMEP and OCN differ in both such that EMEPs estimates for leaf-level ozone concentrations differ from the estimates by OCN. Furthermore leaf-level ozone concentrations are calculated separately for each PFT because the PFTs differ in their LAI values. EMEP and OCN simulate different groups of PFTs and a different number of PFTs. We added
an explanation to the Methods section.

Q: p. 5 Ra is the resistance between the surface (near-canopy) and 45 meter (i.e. it is not at a level of 45m).
A: Changed to 'between 45 m height and the canopy'.

Q: p. 4 l. 18 Can something be said on how this conductance is distributed over the canopy layers- in general how vertical canopy structure is expected to influence ozone uptake
A: Leaf N is generally highest in the top canopy and monotonically decreases with increasing canopy depth. Following this stomatal conductance and $O_3$ uptake is highest in the upper canopy and lowest in the bottom of the canopy. Included in Methods section.

Q: p. 5 l. 19: was this calibration necessary for this study, or more general for OCN model results?
A: This calibration is generally necessary to yield reasonable conductance values in OCN.

Q: p. 6 mention which three PFTs were considered for this LAI+1 approach? Probably for the readers not wanting to go back to older papers, a table listing some characteristics of the PFTs (appendix) would be useful
A: The LAI+1 approach is applied for all tree PFTS (woody PFTs).

Q: p. 6/7 eq. 8,9,10 to what extent are these equations based on observations, or merely model assumptions (and what is the associated uncertainty)
A: These equations are largely the same as used in the EMEP model. As described in

[Figure]

Simpson et al, 2012, Eqn (8), for $F_T$ is taken from Zhang et al. (2012), Eqn (9), $R_{inc}$ is from Erisman et al (1994), and Eqn (10), giving the effect of snow on $R_{gs}$, is also loosely based upon Zhang et al.

Although all such equations are uncertain (all depositions schemes are!), the EMEP model's deposition scheme (and associated D$O_3$SE module for $O_3$) has undergone extensive review and comparison measurements, see for example:

- Emberson, s. D.; Büker, P. & Ashmore, M. R. Assessing the risk caused by ground level ozone to European forest trees: A case study in pine, beech and oak across different climate regions Environ. Poll., 2007, 147, 454-466

- Emberson, L.; Ashmore, M.; Simpson, D.; Tuovinen, J.-P. & Cambridge, H. Modelling and mapping ozone deposition in Europe Water, Air and Soil Pollution, 2001, 130, 577-582

- Emberson, L.; Wieser, G. & Ashmore, M. Modelling of stomatal conductance and ozone flux of Norway spruce: comparison with field data Environ. Poll., 2000, 109, 393-402

- Klingberg, J.; Danielsson, H.; Simpson, D. & Pleijel, H Comparison of modelled and measured ozone concentrations and meteorology for a site in south-west Sweden: Implications for ozone uptake calculations Environ. Poll., 2008, 115, 99-111

- Simpson, D.; Tuovinen, J.-P.; Emberson, L. & Ashmore, M. Characteristics of an ozone deposition module II: sensitivity analysis Water, Air and Soil Pollution, 2003, 143, 123-137

- Simpson, D.; Tuovinen, J.-P.; Emberson, L. & Ashmore, M. Characteristics of an ozone deposition module Water, Air and Soil Pollution: Focus, 2001, 1, 253-262

[Figure]

- Tuovinen, J.-P.; Simpson, D.; Mikkelsen, T.; Emberson, L. D.; Ashmore, M. R.; Aurela, M.; Cambridge, H. M.; Hovmand, M. F.; Jensen, N. O.; Laurila, T.; Pilegaard, K. & Ro-Poulsen, H. Comparisons of measured and modelled ozone deposition to forests in Northern Europe Water, Air and Soil Pollution: Focus, 2001, 1, 263-274

- Tuovinen, J.-P.; Emberson, L. & Simpson, D. Modelling ozone fluxes to forests for risk assessment: status and prospects Annals of Forest Science, 2009, 66, 401

- Tuovinen, J.-P.; Ashmore, M.; Emberson, L. & Simpson, D. Testing and improving the EMEP ozone deposition module Atmos. Environ., 2004, 38, 2373-2385

- Tuovinen, J.-P.; Emberson, L. & Simpson, D. Modelling ozone fluxes to forests for risk assessment: status and prospects Annals of Forest Science, 2009, 66, 401

In any case, it can be noted that the low-temperature and snow terms given by Eqns (8) and (10) are only really important in conditions for which ozone uptake will be very small.

Q: p. 7 l.18 1 mmol/m2: is this referring to cumulative ozone uptake? Is this published (reference?). I am not sure if such sensitivity with an atmospheric model which would include chemistry feedbacks, can be translated into such small uncertainty for the vegetation $O_3$ uptake. Note that there is in general a quite large difference in PBL mixing in a variety of atmospheric models- which in itself already suggests a large uncertainty.
A: Yes, the 1 mmol/m2 is for CUO with threshold 1.6. This estimate comes from tests done for this paper, by running the EMEP model with different assumptions, but it only represented the uncertainty due to the OCN simplifications in resistance terms, not of course the overall uncertainty in the model system. In any case, since we now use a zero threshold, and have modified the OCN resistance terms, a new calculation was

needed.

The respective paragraph was changed to: 'However, a series of calculations with the full EMEP model have shown that the uncertainties associated with these simplifications are small, typically 0.5 - 5 $\mathrm{mmol\,m^{-2}}$. As base-case values of POD0 are typically ca. 30-50 in EU regions, these approximations do not seem to be a major cause of error, at least in regions with substantial ozone (and carbon) uptake. The coupling of OCN to a CTM would be desirable to eliminate this bias.'

Q: p. 8 l-1-20 See remarks above- need to get a better description if/how detoxification is included

A: The flux threshold simulating detoxification was skipped and all simulations were rerun.

Q: p. 8 l. 8: do I understand correct that in the rest of the text $CUO_5^{1.6}$ would refer to equation 15; while CUO would refer to use Fst in equation 15. This needs to become clear- and the correct equations need to be given.

A: The flux threshold was skipped and following this also $CUO_5^{1.6}$ was skipped. The cumulated $O_3$ uptake (CUO) derives from the accumulation of the ozone uptake without any flux threshold ($F_{stC}$).

Q: p. 9 l. 5. Would a sensible variation of dl (equation 16) also be a critical parameter?

A: The objective was to test functionality of the implemented deposition scheme. The validity of the implemented damage function is a very interesting topic however would have expanded the manuscript too much. We are currently working on evaluating different damage functions implemented in OCN in their ability to reproduce observe damage relationships.

Q: How was this subset of parameters selected.

A: Key parameters of the deposition scheme which determine leaf-level $O_3$ concentrations and hence $O_3$ uptake are investigated. The variable $R_b$ is also a key variable of the deposition scheme and was added to Fig. 3.

Q: p. 9 l. 22 What is the La Thuille dataset?
A: The La Thuile Dataset contains the data of all sites and years of the FLUXNET network. The respective web link is included as a reference: 'http://fluxnet.fluxdata.org/data/la-thuile-dataset/'.

Q: p. 9 l. 25 How many years were need to reach equilibrium?
A: 1200 simulation years for the vegetation and 12000 years for the soil secure equilibrium.

Q: What was the criterium for equilibrium?
A: Equilibrium is reached when the carbon and nitrogen pools in vegetation and soil show no trend anymore as mentioned on p 9 l 25 (of the manuscript in discussion).

Q: p. 9 l 28: Which EMEP simulations were providing this 100 years transient concentrations? Is there a reference?
A: p 9 l 25 indicates that more details regarding EMEP are given in section 2.5, including also a reference.

Q: p. 9 l. 29 Appendix tab 1? I think just table 1
A: The appendix section was unintentionally included into the main part of the paper.

Q: p. 10 I understand that the purpose of section 2.4 is to derive trust in the model,

when testing to observable parameters. I would however need some more insight in why morning/evening fluxes need to removed, and data with the different soil moisture. What would be the effect of not removing such data?

A: The morning and evening hours are removed since in this time dew condensation on the leaves causes a wet canopy. This causes an alteration in latent heat exchange ($LE$) such that FLUXNET observed canopy conductance, which is inferred from $LE$, is prone to a high uncertainty in these times. Soil moisture constraints directly impact the simulated net photosynthesis (see $\Theta$ in Eq. 5). It is hard for a global model, not tuned for the specific site, to properly model the drying of the soil and onset of soil moisture stress (which depends e.g. on soil type and texture as well as the degree of root penetration). By excluding data under soil moisture stress this bias is removed.

Q: p. 10 l. 29- brought into equilibrium. How done?

A: The model is run with the 1961-1970 forcing until equilibrium of the carbon and nitrogen pools in vegetation and soil is reached. The forcing for each year of the Spinup phase is randomly chosen from the period 1961-1970. Changed in the text from 'with 1961-1970 forcing' to 'by randomly iterating the forcing from the period 1961-1970'.

Q: p. 11 figure 1: obviously the largest discrepancy is found for LAI and in p. 12/l. 18 the authors suggest that this is not important. How is it possible to have realistic GPP etc; and such a spread in LAI? Please explain

A: The LAI measurements presented here are point measurements of years outside the simulation period. The actual LAI values at the FLUXNET sites during the simulation period might be different. Furthermore in OCN GPP depends on LAI in a non-linear relationship where GPP saturates with increasing values of LAI (saturation point at a LAI of approximately 4). When LAI increases further the lower canopy does not get sufficient light to increase GPP. GPP however is not only determined by LAI,

but also e.g by temperature, radiation and soil moisture stress what might ameliorate differences in LAI. Added: 'Modelled GPP does not only depend on LAI, but also on light availability, temperature and soil moisture.'

Q: p. 12 l. 24. While it is facilitating the discussion to focus on only 3 stations, some words on how representative these stations were for others would be welcome.
A: The three sites were chosen to be examples of the 3 major categories. The respective sentence was reformulated to: 'For further evaluation of the modelled ozone uptake, we analysed the diurnal cycles at three sites, one of the three categories broadleaved, needle-leaved and C3 grass sites respectively.'

Q: p. 13 l. 5-25 I would advise to also see Hardacre, Atmos. Chem. Phys., 15, 6419-6436, 2015, for further opportunities to evaluate the ozone deposition velocities and fluxes.
A: Hardacre et al. 2015 was included into the evaluation of the deposition velocities. The fluxes given in Hardacre et al. 2015 are total dry deposition values. In the manuscript here we evaluate the stomatal fraction of the dry deposition ($F_{stC}$). A comparison of both is not possible.

Q: p. 13 l. 35 Can you confirm that sapflow measurements are not reliable for this study?
A: We can not judge which measurements (eddy covariance or sap flow) are reliable. However we observe that between both techniques the estimates of canopy conductance differ by a factor of more than 10 and that our estimates reported here are more similar to estimates done by measurements conducted by the eddy covariance technique. Since canopy conductance drives $O_3$ uptake, a 10 fold higher canopy conductance results in an approximately 10 fold higher $O_3$ uptake rate (disregarding in this approximation the feed back of $O_3$ uptake into the leaf on decreasing leaf-level $O_3$

concentrations).

Q: p. 14 repeat here that Fr is the ratio of stomatal to overall flux. It would be interesting to give average values (per ecosystem/PFT) over the months. Perhaps for an appendix? I think this could be useful for comparison in future studies.
A: The explanation of Fr on p 14 was changed from 'The ratio between the vegetation ozone uptake and the total surface uptake (Fr)' to 'The ratio between the stomatal ozone uptake and the total surface uptake (Fr)'. A graph showing monthly mean values of key ozone metrics is added to the appendix (Appendix 11).

Q: p. 14 l. 34 I didn't quite understand the sentence not zero because accumulate over several years. Isn't it simply that there is already some photosynthesis activity?
A: Accumulated ozone is shed when leaves are shed. Deciduous PFT's shed all accumulated $O_3$ at the end of the growing season when the leaves are shed. Evergreen species only shed a fraction of their leaves and keep the leaves that have already taken up $O_3$ for several years. The CUO decreases in winter when the evergreens shed part of their leaves but since they do not shed all the CUO remains greater than zero.

Q: p. 15 l. 8-10. It is not clear to me whether OCN has croplands, and if so what crop? The authors mention C3 crops- I guess that would be mainly wheat?
A: OCN simulates 12 PFTs including 8 tree PFTs, 2 grass PFTs and 2 crop PFTs. The crop PFTs are a generic C3-crop and a generic C4-crop. As species are not explicitly simulated for the tree and grass PFTs this is also not done for the crops.

Q: p. 15 l. 17 Figure 6a is an EMEP model output?
A: Ozone concentration plotted in Fig. 6a is the forcing OCN uses for the simulations.

[Figure]

This forcing is provided by EMEP.

Q: p. 15 Appendix 12 ab missing. Do you mean Figure 12? See=>sea
A: The appendix section was unintentional included into the main part of the paper (Appendix 12 == Fig. 12).

Q: p. 15 l. 34 interesting dynamical/phonological feedback, but it also reminds that things like early senescence are probably not included.
A: Yes, early senescence is not included.

Q: p. 16 section 3.5 Please remind reader of what D-STO and ATM were? See section 2.6.
A: Changed to 'the D-STO model (non-stomatal depletion of ozone is zero) and 20-25% for the ATM model version (canopy $O_3$ concentration is equal to the atmospheric concentration) '.

Q: Appendix 11 and 13 are missing.
A: The appendix section was unintentional included into the main part of the paper.

Q:L. 13 uptake and accumulated: rephrase in:accumulated uptake.
A: Done.

Q: p. 16 spell out the meaning (remind the reader) of CUO1.5 and 5.
A: Due the omission of the flux threshold $CUO_5^{1.6}$ does no longer occur.

Q: p. 17 section 4.1, l..10 Interactions with VOCs (as well as soil NOx emissions,

see Ganzeveld's paper), are important. But I don't understand how they are implicitly included, especially in the OCN framework.

A: All ozone deposition models that we are aware of have terms for the stomatal uptake of $O_3$, and then for 'non-stomatal' terms in some form ($G_{ns}$ in Eqn. 4). The stomatal terms can be estimated quite well, e.g. from water fluxes. Unfortunately, the values assigned to $G_{ns}$ cannot be determined from first principles or even experiment because of the complexities of the surface characteristics (moisture films, chemical compounds on leaves, etc, Fowler 2009), and of interpreting flux measurements in the chemically-active conditions associated with vegetation canopies. Thus, the $G_{ns}$ terms encompass both deposition and chemical processes - they are essentially tuned to give reasonable values for deposition velocities across diurnal cycles for example.

Q: p. 17 l. 20: was $O_3$ needed to reach this good agreement. Probably not- explain.

A: Given the uncertainty of the observations and model results the inclusion of ozone damage does not improve the fit of the model results to the observations. The comparison to FLUXNET data was mainly meant to show that the model in general produces realistic values especially for the canopy conductance ($G_c$), since $G_c$ is a major factor determining ozone uptake and hence estimated damage.

Q: p. 19 l 3. As explained above, I think this warrants some additional analysis.

A: The validity of the implemented damage function is a very interesting topic however would have expanded the manuscript too much. We are currently working on evaluating different damage functions implemented in OCN in their ability to reproduce observe damage relation ships. This is a topic of its own.

Q: p. 19 l. 22 impacted=>determined?

A: Changed to determined.

**3 Figures**

Q: Figure 6: Why are the units of panels b and c different?
A: The units are different because different variables are plotted. In panel b the mean ozone uptake rate [$\mathrm{nmol\,m^{-2}\,s^{-1}}$] is plotted. In panel c the mean ozone accumulation [$\mathrm{mmol\,m^{-2}}$].

Q: The chosen range doesn't work well for panel b (all purple).
A: The color range is not the problem in the big purple area. The values of the mean uptake rate all lie between 1.9 and 2, which simply is a small range.

Q: Figure 7: it is hard to discriminate the colors in Figure 7.
A: The color palette is changed from rainbow to restricted color gradients (palettes from ColorBrewer 2.0).

Q: Figure 9: legenda describing a) can be improved.
A: Changed from 'no ozone deposition scheme (ATM),' to: 'canopy $O_3$ concentration is equal to the atmospheric concentration (ATM)'

Q: Figure 12: color scheme doesn't work (mostly red)- more resolution for low values is need (0-10%). For C4 crops- is irrigation considered?
A: Irrigation is not considered for crops. The graph is skipped due to it's minor value in explaining observed results.

---

## Author Comment (AC2) · 9 Nov 2016

Dear Referee,

we thank you for your detailed and constructive comments that helped considerably to improve the manuscript.

Yours Sincerely
Martina Franz

[Figure]

**1 General comments**

Q: The authors use "ozone" and "O3" fairly randomly throughout the manuscript. I would suggest sticking with one or the other.
A: 'Ozone' and '$O_3$' are not used randomly. '$O_3$' is used when we refer to the chemical substance and 'ozone' is used when we refer to the damage $O_3$ causes or the included deposition scheme. In the cases where this was not consistent we changed it to the above mentioned rule. We would like to keep it that way if it is not distracting.

**1.1 Abstract**

Q: P1, L6 - This is the first use of the acronym OCN - please explain what it is.
A: Added: '(the OCN terrestrial biosphere model)'

Q: P1, L12 - "update" should read "uptake"
A: Done.

Q: P1, L15-6 - Please re-word, this is hard to follow. I think that you are saying: "When applied at the European scale, we find that including our new ozone deposition scheme substantially affects simulated ozone"
A: Changed to: 'When applied at the European scale, we find that the inclusion of the deposition scheme substantially affects simulated ozone ...'

**1.2   Introduction**

Q: P2, L22 - replace "consequence" with "result"
A: Done.

Q: P2, L24 - replace "extend" with "extent"
A: Done.

Q: P2, L27-29 - I suggest making the point here that AOT40 is currently used for regulatory assessment purposes in Europe.
A: Changed from 'A widely used example' to 'The initial standard tool' And furthermore is added: 'Observed ozone damage in the field seems to be better correlated to flux-based risk assessment compared to concentration based methods (Mills et al., 2011). Following this the LRTAP Convention recommends flux based methods as the preferred tool for risk assessment (LRTAP Convention, 2010).'

Q: P2, L32-33 - Please could the authors explain what they mean by "regional provenances". Do they mean that the same species in different geographical locations differ? Or that different regions have different ecosystems?
A: It is meant that canopy conductance of the same species differs when grown in different geographical locations as well as differences exist between species. Changed to: 'A significant caveat of concentration-based assessments of ozone toxicity effects is that species differ vastly in their canopy conductance as well as regional provenances of one species.'

Q: P3, L8 - Up until this point the authors have referred to AOTX. As AOT40 is the regulatory metric and one that they use in subsequent analysis and discussion I suggest they clearly define AOT40 at this point.

A: '(AOTX above a threshold of 40 ppb)' is added.

Q: P3, L23 - I suggest the authors make the point that the threshold values are species-specific to account for plant sensitivity/tolerance to ozone.
A: ', depending on the specific species sensitivity to ozone. ' is added to the sentence.

**1.3 Methods**

Q: P4, L20 - The model acronym EMEP MSC-W should be defined here rather than at the end of the paragraph, e.g. "The ozone and N-deposition data used for this study are provided by the EMEP MSC-W (European Monitoring and Evaluation Programme Meteorological Synthesising Centre - West) chemical transport model (CTM; Simpson et al., 2012a)."
A: Done as suggested.

Q: P4, L22 - insert "been" between "have" and "documented"
A: Done.

Q: P5, L1 - replace "in" with "at" and remove "height"
A: Done.

Q: P5, L7 - replace "in" with "at" and remove "height"
A: Changed to 'between 45 ᴍ height and the canopy' according to F. Dentener's comment.

Q: P5, L15-6 - replace "leafs internal" with "internal leaf"

A: Done.

Q: P5, L16 - parentheses should only be around "2005"
A: Done.

Q: P5, L17 - replace "ozone to water vapour" with "ozone from water vapour"
A: Done.

Q: P5, L19 - is this factor of 0.7 included in Zaehle and Friend or is this new for this current study?
A: It is new in this study. Yet this calibration is generally necessary to yield reasonable conductance values in OCN.

Q: P6, L11 - please explain more clearly what is meant by a low temperature correction factor and why it is needed.
A: According to Simpson et al. (2012) and Zhang et al. (2003) FT is needed since at temperatures below $-1\,°C$, non-stomatal resistances increase up to two times (hence also the boundary of $1 \leq F_T \leq 2$). Added: For temperatures below -1 °C non-stomatal resistances are increased up to two times (Simpson et al., 2012; Zhang et al., 2003)..

Q: P6, L11 - suggest rewording to: "is scaled by a low temperature correction factor, FT, such that"
A: Changed to: 'is scaled by a low temperature correction factor $F_T$ and'

Q: P6, L13 - suggest rewording to: "where TS is the 2m air temperature (C; Simpson et al., 2012a, eq. 60) and 1<FT<2."

A: The reference by Simpson et al. (2012) also refers to $1 \leq F_T \leq 2$, hence the proposed alteration would take away information.

Q: P6, L20 - replace "Like" with "As"
A: Done.

Q: P7, L1 - parentheses should only be around "2003"
A: Done.

Q: P7, L4 - suggest combining to give: "0.5, to prevent negative values in the first fraction of eq. 10".
A: Done.

Q: P8, L4 - Why PODl? My understanding of PODY is that the Y stands for the threshold value not the canopy level.
A: Yes. The PODY usually refers to the top canopy layer and not the canopy integrated value contrary to CUO. The 'l' was there to indicate the same canopy layer as in CUO, however I also see that it is misleading. I erased the 'l'.

Q: P8, L13 - What is the physical (real-world) interpretation of the parameters 0.22 and 6.16 in eq. 16?
A: The parameter 6.16 suggests that at zero ozone uptake net photosynthesis is damaged by 6.16 %. Per mmol accumulated ozone uptake the net photosynthesis is further damaged by 0.22 %.

Q: P8, L13-4 - Why not just divide by 100 in the equation itself?

A: The equation in the numerator is the original equation by Wittig et al. (2007) which gives the damage in percent. Since we needed the fraction [0,1] instead of the percentage it seemed the clearest way to indicate this.

Q: P8, L17 - Please explain to the general audience why a reduction in An results in reductions in Gst and (particularly) Ci. It is not intuitive why this would reduce internal concentrations.
A: The stated reduction of $C_i$ was wrong. ' and $C_i$' was erased.

Q: P8, L23 - parentheses should only be around "2010"
A: Done.

Q: P9, L2-3 and throughout - I would suggest that the authors re-define or at least use a word description each time these parameters are re-introduced at the start of a new section; else provide a table listing the key parameters for the reader to refer back to.
A: Are reintroduces again.

Q: P9, L11 - Are the "summer months" defined here the same as what is then referred to as the "growing season"; if so, please make clear, if not, please define growing season separately.
A:Growing season is not equal to summer month. Growing season is defined: 'To derive average growing-season fluxes (bud break to litter fall), ...'

Q: P9, L21 - Please explain what is meant by "site levels". Is this "site-specific" i.e. OCN is run as a column model rather than a 3-D regional model?
A: site level means that the simulation is run only on a single set of coordinates and not for a region. Changed to: 'The site levels simulations (single point simulations) ...'

Q: P9, L22 - square parentheses are not required around CO2 as the text includes the word "concentrations".
A: Parentheses are erased.

Q: P9, L23 - parentheses should only be around "2015".
A: Done.

Q: P9, L23 - rearrange this to read: "Reduced and oxidised nitrogen deposition in wet and dry forms and hourly"
A: Done.

Q: P9, L27 - O3 should be subscript
A: Done.

Q: P9,L28-9 - Why not use GCM output or reanalyses data where there is a lack of observation data?
A: We have observation data for all stations but only for the observation period. The model however needs to be in equilibrium to yield sensible results hence a Spinup has to be run (approximately 1200 years for the vegetation). To be able to use the GCM climate it would have to be bias corrected for all climate variables to prevent a step change when changing to use the observed data at the FLUXNET stations for the observation period. This bias correction is much work besides the fact that bias correction except of temperature is not trivial. The use of the observed climate for the Spinup period constitutes a secure way to prevent step changes at the start of the observation period.

Q: P9, L30 - what do the authors mean by time-varying here? Surely the progressive simulations also used data that varied with time. Do the authors mean that here it is observations from the site in question for the years in question?

A: Meant is the year in question. Rephrased to: 'The observation years (see Appendix Tab. 1) are simulated with the climate and atmospheric conditions (N deposition, $CO_2$ and $O_3$ concentrations) of the respective years.'

Q: P10, L2 - Why have the authors chosen to base LAI on single point, time-specific observations rather than e.g. MODIS LAI data? It seems that this introduces a considerable source of uncertainty.

A: MODIS data are also subject to a considerable amount of uncertainty. Furthermore the resolution of MODIS data is an additional source of uncertainty. Using observation directly from the site in question seemed to be the most reliable source.

Q: P10, L5 - parentheses should only be around "2015"

A: Done.

Q: P10, L6-7 - suggest rewording to read: " are filtered prior to deriving average growing-season fluxes to reduce the effect of model biases on the model-data comparison. Night-time and "

A: Done.

Q: P10, L9 - please explain what a "modelled soil moisture constraint factor" is, and why a threshold of 0.8 has been chosen as a filter. Is this based on observations suggesting severe drought impacts alter fundamental plant functioning?

A: The soil moisture constraint factor is the $\Theta$ in Eq. 5. It constrains net photosynthesis when soil moisture decreases and takes values between zero and one. The threshold

of 0.8 secures relative humid soils since site specific soil moisture constraints are hard to capture with a global model. The drying of soils is hard to capture for a model operating on 1 degree resolution since it depends e.g. on soil type and texture as well as the degree of root penetration). By excluding data under soil moisture stress this bias is removed.

Q: P10, L10-1 - suggest rewording to "Daily mean values are calculated from the remaining time steps only where both modelled "
A: Done.

Q: P10, L14 - why only use July here when the rest of the analysis is conducted for JJA?
A: Only one month (July) was chosen since it is easier to compare means of one month to reported values in the literature than mean values over several months.

Q: P10, L14-15 - why not use the same light level to define daylight as you used to filter the data previously?
A: For the hourly mean values the threshold of 100 $Wm^{-2}$ is used to have a sharp cut-off of values with small light intensities where photosynthesis is little active and dew might bias the estimated $G_c$ of FLUXNET. To calculate daily mean values such a restrictive boundary is not necessary since the early morning hours are not investigated separately.

Q: P10, L16 - suggest rewording to "..FR and for both modelled and FLUXNET-observed GPP"
A: Done.

Q: P10, L22-3 - suggest rewording to "1999). Reduced and oxidised nitrogen deposition in wet and dry forms and ozone"
A: Done.

Q: P10, L25 - parentheses should only be around "2014b"
A: Done.

Q: P10, L25 - insert "and are" before"scaled back"
A: Done.

Q: P10, L27 - parentheses should only be around "2011"
A: Done.

Q: P10, L28 - square parentheses are not needed around CO2.
A: Skipped.

Q: P10, L28 - parentheses should only be around "2015"
A: Done.

Q: P10, L29-30 - Please check dates. If 1961-1970 is used as a spin-up shouldn't the simulation then start at either 1961 (repeating the first 10 years) or from 1971?
A: The transient simulation starts at 1961 and ends in 2011 (since the MTE period extends to 2011). Changed to '1961-2011'.

Q: P10, L32 - Please explain what an MTE product is.
A: MTE is a machine learning technique. Changed to: 'An up-scaled FLUXNET-MTE-

product of GPP (Jung et al., 2011), using the machine learning technique: model tree ensembles (MTE),'

Q: P11, L2 - replace "Different" by "In contrast"
A: Done.

Q: P11, L3 - O3 should be subscript
A: Done.

Q: P11, L3-4 - Please explain for the non-specialist audience why the resistances result in a lower canopy concentration.
A: Changed to: 'Due to these resistances, the deposition of ozone to leaf-level is reduced, and the canopy $O_3$ concentration is lower than the atmospheric $O_3$ concentration.'

**1.4 Results**

Q: P11, L11 - what do the authors mean that they agree "within the standard deviations"? Are they stating that the data overlap? It would be better to demonstrate this goodness of fit with robust statistical analysis.
A: 'within the standard deviation' is substituted by 'well'. A table reporting the: 'Coefficient of determination ($R^2$) and Root Mean Square Error (RMSE) for $GPP$, canopy conductance ($G_c$), and latent heat fluxes ($LE$) for all sites, sites dominated by broadleaved trees, needle-leaved trees, C3 grass, and C3 grass except of the AT-Neu site (outlier).' is added to the Appendix and cited in section 3.1 Evaluation against daily eddy-covariance data '(see Appendix Tab. 2 for $R^2$ and RMSE values)'. Given

the observational uncertainty, the model performance appears to be acceptable.

Q: P11, L13 - should read "very close, with only slight under-"
A: Changed.

Q: P12, L3 - remove extra ")" after 10 a
A: Done.

Q: P12, L5-6 - please give an example of site management that might result in such variability
A: Mowing can change LAI strongly and through this impact estimated GPP and $G_c$. '(e.g. mowing)' is added.

Q: P12, L8 - why should LE be overestimated and GPP underestimated by OCN at broadleaved forest sites?
A: We can only speculate that a bias in the estimation of the FLUXNET LE might be the cause for this. It might also be possible that the observed water use efficiency (WUE) is not properly captured by OCN, what however seems unlikely to be the major reason since $GPP$ and $G_c$ do not show such a bias when compared to each other.

Q: P12, L13 - what do the authors mean by "vary more widely"? Do they mean that there is a greater difference between modelled and measured values or that there is greater variability in the differences?
A: There is greater difference between modelled and measured values compared to the needle-leaved tree sites mentioned in the preceding sentence.

Q: P12, L14 - Do the means still lie within one standard deviation or not? Is there a tendency for the model to consistently under- or overestimate?

A: Changed to 'The modelled $G_c$ at sites dominated by C3 grasses is in very good agreement to FLUXNET $G_c$ with slightly overestimating $G_c$ at 2 out of 3 sites except for the DE-Meh site, where means differ outside the standard deviation (see Appendix Fig. 10 b).'

Q: P12, L15-22 - move to SI

A: We would like to keep this paragraph included (like Referee 1), however we can move it to SI if demanded.

Q: P12, L23 - general comment regarding section 3.2: Do the reported "biases" in the diurnal cycles reflect those of the means? i.e. is GPP underestimated at the broadleaf site.

A: The biases are partly reflected by the hourly value. For instance the fact that the needle-leaved trees site matches observed values best. For the broadleaved trees GPP shows a bias towards underestimation by the daily mean values, however is overestimated on the site shown for the hourly values. The Gc shows a slight bias towards overestimated by the mean and is also overestimated by hourly values. There seems to be little benefit for the readers gain of knowledge to compare the exemplary site to the bias of the category so much in detail. A sentence to compare the general pattern of daily means and hourly values is added: 'Similar to the daily mean values (see Fig. 1 a,b) the mean hourly values show the best match of $GPP$ and $G_c$ for the needle-leaved tree site and stronger deviations for the sites covered by broadleaved trees and C3 grasses.'

Q: P12, L24 - diurnal profiles of which variables? State here

A: Done.

Q: P12, L32 - remove unnecessary parentheses after m and n.
A: Done.

Q: P12, L32 - should read: "with particularly good agreement"
A: Changed.

Q: P12, L32 - surely it's more relevant that it is an evergreen needle-leaf forest that it is Finnish?
A: Changed to 'needle-leaved site'.

Q: P12, L34 - again, state the type of landcover at this site
A: 'Italian' substituted by 'grassland'.

Q: P13, L1 - Again please explain what is meant by the means being within the standard deviation.
A: Changed to: 'The modelled hourly values fall in the range of the observed values.'

Q: P13, L2 - The maximum variability at CH-Oe1 seems to occur during the middle of the day
A: Yes, this fact was erased and changed. Changed from 'where the observed values became highly variable. ' to 'where the observed values increase again.'

Q: P13, L3 - "whereas" is all one word
A: Changed.

Q: P13, L4 - what about the peak GC at the CH-Oe1 site? Is it also overestimated by the model?
A: Yes. Respective sentence changed to: 'and overestimates peak $G_c$ at the CH-Oe1 site.'

Q: P13, L5 - "simulate" rather than "simulated"
A: Changed.

Q: P13, L5-6 - is this not a serious short-coming of the model water response parameterisation? I thought the midday depression in GC was a well observed response to water stress. Please comment on the likely implications for your results and conclusions?
A: The midday depression of $G_c$ is a well observed phenomena and ought to be captured better by the model. However how strong the midday depression is and if it occurs at all is species and site specific. It does not occur for instance at the FI-Hyy site. The IT-Ro1 site shows that the model is at least in some cases able to capture important patterns like the midday depression of $G_c$. OCN however is a global model and not especially tuned for the specific sites such that the features of some sites will be captured better than others. Furthermore the observations at the CH-Oe1 site show very wide error bars, which also indicates the uncertainty in the observations! In times when $G_c$ is underestimated the ozone uptake will also be underestimated and will result in a lower estimated damage. However since it is not a general pattern that the midday dip is not reproduced, this seems not to have a strong impact on the reported results and conclusion. One has to keep in mind that the modelling of ozone damage underlies many uncertainties as well as the observations against which the modelling results are evaluated.

Q: P13, L7-> Please either change the order of the panels in Figure 2 or the order of

the text so that you are presenting the results of the panels in the order in which they appear.

A: Order in the text is changed.

Q: P13, L9- 15 - How is RC measured? or is it back-calculated from observed ET and LE? Please comment on the reliability of the observations.

A: $R_c$ can be inferred from measurements by the eddy covariance technique (Coyle et al., 2009; Gerosa et al., 2004; Mikkelsen et al., 2004). The total deposition of ozone is calculated from the ozone concentration at measurement height and the fluxes measured by the eddy covariance technique (total ozone deposition). $R_c$ can be inferred from the total deposition as the residual when subtracting $R_a$ and $R_b$. Eddy covariance measurements and derived flux and conductance estimates are subject to a diverse set of random and systematic errors (Richardson et al., 2012). A lack of energy balance closure can cause underestimation of sensible and latent heat as well as an overestimation of available energy, with a mean bias of 20 % where the imbalance is greatest during nocturnal periods (Wilson et al., 2002). Since $R_c$ is inferred from measured fluxes the calculation of $R_c$ underlies the uncertainties of the flux measurements.

Q: P13, L9-15 - what are the implications of the model deviations from observations?

A: The main purpose of this evaluation is to show that our model produces realistic values for key ozone variables. The modelled values are within the range of observed values and show the expected diurnal pattern. Deviations from the values reported in the literature are expected since we neither model the specific sites nor the species. That also means that also the climate and ozone concentrations of the observations can not be reproduced by OCN which both have a major impact on the modelled ozone variables. Since the modelled values are within the observed range reported in the literature it can be assumed that our model works fairly well.

Q: P13, L15 - should read " observed which is slightly lower"
A: Changed.

Q: P13, L16 - the minimum velocities appear to be lower than this value for crops
A: I am not sure what 'for crops' refers to, since we do not model crops here. In case it is meant that for the CH-Oe1 (grassland site) site minimum $V_g$ is lower than 0.002: Yes, it is approximately 0.0015 for CH-Oe1, however I think it is 'approximately 0.002 $\mathrm{m\,s^{-1}}$' when stating the mean minimum $V_g$ for all three sites.

Q: P13, L18 - "barely" should read "barley"
A: Changed.

Q: P13, L16-20 - The modelled velocities at your crop site are well below these.
A: We do not model a crop site, the CH-Oe1 site is a grassland site. The crop values only indicate the observed range, since trees might also not be the best choice to compare with. Besides our modelled peak values of $V_g$ are approximately 0.0055 $\mathrm{m\,s^{-1}}$ which in our notion compares well to observed ranges of 0.003-0.009 $\mathrm{m\,s^{-1}}$ at noon (Gerosa et al. 2004) for a barley field and approximately 0.006 $\mathrm{m\,s^{-1}}$ at noon for a wheat field (Tuovinen et al., 2004).

Q: P13, L20 - please rephrase to "The estimates for Hyytiälä also agree"
A: Changed to: 'The estimates for FI-Hyy also agree'.

Q: P13, L16-23 - It would be helpful if you compared the data site by site as before
A: This is done here, only that we start with the CH-Oe1 site, followed by FI-Hyy and last IT-Ro1. The reason for evaluating IT-Ro1 last is that for broadleaved trees we found only daily mean values to compare with, such that the actual diurnal cycle can not be properly evaluated. Hence it seems better not to start with this site.

Q: P13, L23 - Why is Vg so noisy for IT-Ro1?

A: $V_g$ is determined by the total ozone uptake which is composed of a stomatal and a non-stomatal fraction. The noise in the stomatal component of the total uptake ($F_{stC}$) causes the noise in $V_g$. $F_{stC}$ is determined by $G_c$ and leave-level ozone concentrations. Since $G_c$ shows not much noise it can be assumed that the day to day variability of the leave-level ozone concentration is the cause of the noise in $F_{stC}$ and $V_g$.

Q: P13, L24 - Perhaps it is worth making the point that Vg is not zero because of non-stomatal deposition.

A: The sentence on P13, L24 deals not anymore with $V_g$ but with $F_{stC}$. And $F_{stC}$ is not zero during the night since a minimum conductance occur also during the night even though photosynthesis is zero.

Q: P13, L27-28 - Why is there such large variability in the afternoon at IT- Ro1? Is that another sign of water stress?

A: As already mentioned above: $F_{stC}$ is determined by $G_c$ and leave-level ozone concentrations. Since $G_c$ shows not much noise it can be assumed that the day to day variability of the leave-level ozone concentration is the cause of the noise in $F_{stC}$.

Q: P12-13 - general comments: For Rc, Vg, FR, FStC: what are typical/expected profiles of these variables? Do we really only have observations at 1 or 2 times per day with which to assess model skill? How do these output data compare with estimates from other models? I would strongly recommend that much of the content here is moved to SI and/or presented in a table, with this section only highlighting a few key or interesting features.

**BGD**

A: The expected diurnal profiles are as modelled by OCN, with peak value during the day for all variables except of $R_c$ where maximum values are expected during the night. Hence the diurnal pattern is modelled appropriately. Certainly there are observations that do report on a high temporal resolution (Mikkelsen et al., 2004; Gerosa et al., 2004, 2003). However, we do not model the sites where the observations are conducted, it does thus not seem appropriate to compare details of model and data, especially since differences between species are high (see ranges of cited values for different species). In our notion it is interesting to show the diurnal pattern including the hourly standard deviation. It seems important to show that the diurnal pattern of the variables can be reproduced by the model and how this varies between the sites. Information about when standard deviation is typically high or low and how and why it is high for some variables would be skipped when condensing Fig. 2 into a table. For instance the fact that the high noise level for $F_{stC}$ at IT-Ro1 can not be explain by noise in $G_c$ is information that would get missing.

Q: P14, L2 - add a reminder in the parentheses that GCO3=GC/1.51
A: Changed to: '$G_c^{O_3} = \frac{G_c}{1.51}$'

Q: P14, L3 - Is this ratio essentially the proportion of deposition that is stomatal?
A: Yes.

Q: P14, L3-9 - Why have the authors chosen to report the 24-hour average for this variable and not for the others? Section 3.3 This section and the accompanying figure should be moved to SI, with only a few key headline findings included in the main text.
A: The 24-hour average is given for $F_R$ since for instance in Cieslik (2004) the reported flux ratios are mean values (for diverse sites listed in a table) and the daily mean value in our graph should facilitate the comparison with this table. If this 24-hour mean value is a distraction to the reader it can be removed, otherwise we would like to keep it.

The included ozone deposition module is the key component for simulating ozone uptake and damage. Since it is done the first time to include such a detailed deposition model into a global terrestrial biosphere model it seems to be very important to show that this inclusion worked properly. That means that the results are fairly robust against the exact parametrisation (Fig. 4) but also that perturbations in one variable cause expected effects in related/depending variables (Fig. 3). Furthermore it seems quite important to show which variables of the deposition scheme mainly impact the estimated ozone uptake and hence damage (Fig. 3).

Q: P14, L12 - replace "constraint" with "constrained"
A: Changed.

Q: P14, L13 - "boreal" would be a more useful descriptor than "Finnish"
A: Changed.

Q: P14, L13 - replace "except of" with "except for"
A: Done.

Q: P14, L14 - replace "describing" with "which describes"
A: Done.

Q: P14, L17 - replace "compared" with "relative"
A: Done.

Q: P14, L22 - insert "canopy conductance" before "GC"
A: Done.

Q: P14, L23 - replace "what causes" with "resulting in"
A: Done.

Q: P14, L24 - replace "compared"with "relative"
A: Done.

Q: P14, L25 - remove "changed values for"
A: If changed values would be removed it would sound as if only $r_{ext}$ and $G_c$ are important for the fluxes however this is not the case. The message is that $r_{ext}$ and $G_c$ need to be properly modelled because changes in their values impact the modelled fluxes. Thus we would like to keep the sentence unchanged.

Q: P14, L26 - explain the units (%/%)
A: '0.1 (%/%)' is substituted by '0.1 % due to a 1% change in the variables/parameters of the deposition scheme.'

Q:P14, L27 - remove "very" and "varying"
A: 'very' is removed. Varying is not removed since the message is that perturbations (variations) of $r_{ext}$ and $G_c$ little effect $F_R$. It is not the case that $F_R$ is little affected by $r_{ext}$ and $G_c$!

Q: P15, L1-2 - has this phenomena (the effect of needle-shedding on CUO) been evaluated?
A: I am not sure what is meant by 'if the phenomena has been evaluated'. As in our response to reviewer one, we believe that the use of $f_{shed}$ has caused some confusion,

and therefore we have replaced this with $f_{new}$. The CUO itself is only representative of what actually happens in the plant. Ozone does not actually accumulate in the plants. However, CUO is a substitute to estimate potential damage to the leaves/plant. It can be assumed that new grown leaves are healthy. Deciduous plants grow a complete set of new leaves each year and shed all damaged leaf at the end of the growing season and hence also shed the CUO. Evergreen plants keep their leaves for several years but if they would keep accumulating the CUO they would die since damage keep increasing. Hence it is reasonable to assume that if old/damaged leaves are shed also the fraction of CUO they took up will be shed too.

Q: P15, L6-7- what percentage is 250 gC/m2/yr?
A: The range of $\pm$ 250 $\mathrm{g\,C\,m^{-2}\,yr^{-1}}$ is skipped and substituted by the European mean deviation of OCN from MTE, since this seems to be more informative. The respective sentence is rewritten to: 'Simulated mean annual GPP for the years 1982-2011 shows in general good agreement with an independent estimate of GPP based on up scaled eddy-covariance measurements (MTE, see Section 2.5), with OCN on average underestimating GPP by 16 % (European mean).'

Q: P15, L8 - remove "to this acceptable agreement"
A: Done.

Q: P15, L9 Again what percentage is 400 to 900 gC/m2/yr?
A: Added: '(58 % overestimation on average)'

Q: P15, L12-3 - It also makes it difficult to assess the reliability of the model!
A: Yes, because there might be no reliable source to compare with.

Q: P15, L16 - Please explain how N limitation can lead to overestimation of GPP

A: In the North OCN underestimates GPP compared to MTE not overestimates. Changed to : 'North of $60°N$ OCN has the tendency to produce lower estimates of GPP'. The underestimation might result from N limitation.

Q: P15, L20 - Fig. 6d does not show GPP. Should this read Fig. 5a?
A: Yes, changed to 5a.

Q: P15, L23-4 - Is it not to be expected that AOT40 closely follows absolute ozone concentrations?
A: Yes, it is expected and it is good to be able to compare the AOT40 pattern to the CUO pattern.

Q: P15, L26 - replace "averaged" with "ranged from 60 to 120"
A: Changed.

Q: P15, L27 - move "(Fig 7 a)" to between "Europe" and "and"
A: Done.

Q: P15, L28 - "larger" should read "large"
A: Changed.

Q: P15, L28 - does this refer to Fig. 7b?
A: Yes. '(Fig. 7b)' is inserted at the end of the sentence.

Q: P15, L29 - suggest rewording: "with high cover of C4 PFTs, e.g. Black Sea area (see Appendix 12 a,b)."

A: Done. The graph Appendix 12 is skipped due to it's minor value in explaining observed results.

Q: P15, L30-1 -suggest rewording: "where productivity is low and stomatal O3 uptake reduced by low O3 concentrations or drought control on stomatal fluxes respectively."
A: Changed to : 'where productivity is low and stomatal $O_3$ uptake is reduced by e.g. low $O_3$ concentrations or drought control on stomatal fluxes respectively.'

Q: P15, L31-2 - suggest removing the sentence beginning: "Slight increases or strong decreases"
A: We would like to keep the sentence since it puts the displayed outliers, the positive damage, and the strongest fractional damage into context.

Q: P15, L32 - "increases" should read "increase"
A: Changed.

Q: P16, L3 - replace "by" with "of"
A: Changed.

Q: P16,L4 - insert "Fig. " before "7 c"
A: Done.

Q: P16, L4 - insert "of transpiration" after "3-4%"
A: Done, and European changed to Europe.

Q: P16, L4 - remove "to" before "4-6%"

A: Done.

Q: Q: P16, L5 - insert "relative" before "reductions"
A: Done.

Q: P16, L7 - should read "Black Sea"
A: Changed.

Q: P16, L8 - insert "Fig." before "7 d" and replace "They are" with "These are"
A: Done.

Q: P16, L10 - please explain why a reduction in transpiration matters.
A: Changes in transpiration might impact run-off and surface cooling.

Q: P16, L15 - suggest rewording: "CUO1.6 increases more strongly by 35% "
A: This sentence has been removed since the flux threshold and hence $CUO_5^{1.6}$ has been removed.

Q: P16, L18-9 - It seems to me that in this study simulation D is effectively the base case and D-STO and ATM are sensitivity tests. It would therefore make more sense to swap panels a and c in Figure 9. Furthermore, it seems to me that this is the real headline message of this study - that the ozone deposition scheme substantially alters estimates of impacts. this needs far more emphasis (it is currently hidden by the wealth of detail in the rest of this discussion) and Figure 9 should include further panels showing how CUO changes (see below).
A: We put the ATM case first because this is the common approach if no deposition

model is included (base case). The D-STO model here accounts for impacts of stomatal uptake on leaf-level $O_3$ concentration but still does not account for the non-stomatal fraction and can be seen as an intermediate approach. Our standard scheme accounting for both stomatal and non-stomatal uptake on leaf-level $O_3$ concentrations is the one that comes last such that complexity increases from panel a to c. We would like to keep the present order but can change it if it hampers the understanding of the graph. Furthermore in our notion the general pattern of a decrease in CUO from ATM to D-STO and D is easy to observe from the present graph. Additional panels showing the exact values seem to add little gain of knowledge. Thus we would like to not include them.

To highlight the importance of the deposition scheme more we changed in the Abstract: 'When applied at the European scale, we find that the added complexity of the ozone uptake simulation substantially affects simulated ozone uptake ' to 'When applied at the European scale, we find that accounting for stomatal and non-stomatal uptake substantially affects simulated ozone uptake, ...'

Furthermore we incorporate the importance of the deposition scheme into section 4.1 ( Atmosphere-leaf transport).

**2  Discussion**

Q: This section seems redundant. Much of it is either already stated in the Results section or could be moved to form part of a more robust conclusion.

A: We would like to keep the conclusion short stating briefly the main insights from our work. To reduce redundancy between the results and discussion section we shortened the first paragraph in the discussion section as well as the subsection '4.1 Atmosphere-leaf transport', and '4.2 Site-level evaluation'. The subsection '4.3 Regional damage estimates' seems to us little redundant.

Q: P16, L24-5 - replace "with the aim" with "in order to"
A: Done.

Q: P16, L25 - replace "effect to net" to "effect on net"
A: Done.

Q: P16, L25 - remove "the" before "regional"
A: Done.

Q: P16, L28 - replace "assuming" with "the assumption"
A: Respective sentence is omitted.

Q: P16, L28 - replace "would be identical" with "is identical"
A: Respective sentence is omitted.

Q: P16, L29 - replace "in 45m" with "at 45m"
A: Respective sentence is omitted.

Q: P16, L30-1 - suggest rewording: "and deposition variables i.e. calculated ozone uptake"
A: Respective sentence is omitted.

Q: P16, L32 - P17, L2 - suggest rewriting: "Our sensitivity analysis does show that a correct estimate of canopy conductance is crucial for calculating plant ozone uptake. We find that the model produces reasonable estimates"

A: The respective sentences are omitted in this paragraph. The first sentence ('Our sensitivity analysis ...') is included in subsection 4.1.

Q: P17, L2 - replace "a range of" with "some"
A: Done (the respective half sentence is moved to subsection 4.2).

Q: P17, L7-8 - suggest rewriting: "Reliable estimates of surface ozone concentrations are also essential for calculating canopy ozone uptake FstC"
A: Changed to: 'Reliable estimates of surface ozone concentrations – besides a reliable estimate of $G_c$ – are also essential for calculating canopy ozone uptake $(F_{stC})$.'

Q: LP17, 8-9 - suggest rewriting: "airspace due to biogenic volatile organic compounds (BVOCS) emitted by vegetation is (at least partly) implicitly included in the"
A: We would like to skip the respective sentence since after rewriting the discussion it does not fit anymore.

Q: P17, L9-10 - Does this mean there is a degree of double accounting?
A: No. EMEP accounts for BVOCS (to a certain extend) in the calculation of the $O_3$ concentration in 45 $m$ height. OCN to a certain extend accounts for BVOCS in the calculation of the leaf-level $O_3$ concentration.

Q: P17, L11 - suggest "performance" or "efficacy" in place of "functionality"
A: Respective sentence is omitted.

Q: P17, L15 - suggest combining these to form a single sentence: "changes in GC emphasising the importance"

A: Respective sentences are omitted.

Q: P17, L15-16 - How can reliable estimates be obtained?
A: The respective sentence is omitted. It is of course not possible to simulate the true ozone uptake. However when variables determining ozone uptake are simulated in a reasonable range compared to observations one might call also the calculated uptake reliable (considering the uncertainty in both model simulations and observations). It will anyhow still be an estimate.

Q: P17, L18 - replace "indicates"with "indicate"
A: Respective sentence is omitted.

Q: P17, L26 - replace "impose" with "introduce"
A: Done.

Q: P17, L29 - replace "suitable" with "well able"
A: Respective sentence is omitted.

Q: P17, L30 - remove first occurrence of "finding" and replace "encourages" with "supports"
A: The respective sentence does not anymore exist. "encourages" is replaced by "supports" in a similar sentence.

Q: P18, L2 - reword: "Estimates of the regional damage to annual average"
A: Done.

Q: P18, L2 - make clear this is transpiration rather than temperature (I assume)
A: Transpiration is spelled out.

Q: P18, L2-3 - remove "the period of the years"
A: Done.

Q: P18, L3 - replace "lower" with "low" and "previous" with "previously"
A: Changed to 'lower than previously reported'.

Q: P18, L3 - should read "Meta-analyses" and "an 11Âÿ%"
A: Changed.

Q: P18, L6 - should read "Land Model"
A: Changed.

Q: P18, L7 - reword: "..transpiration have been estimated as 5-20 % for Europe and 2.2% globally "
A: Done.

Q: P18, L9 - reword: "plant types. Damage was only related to cumulative ozone uptake for one plant type with a very small slope"
A: Changed.

Q: P18, L9 - please explain the real-world meaning of a small slope.
A: The higher the slope the more damage occurs per accumulated ozone. The respective sentence is extended to: 'with a very small slope and hence little increase

in damage due to increases in cumulative $O_3$ uptake.'

Q: P18, L14 - use "discrepancies" or "differences" rather than "deviations"
A: Changed to 'discrepancies'.

Q: P18, L14-15 - replace "the usage of very different" with "differences in" and then remove "different", "differing" and "non-identical"
A: Done.

Q: P18, L16 - replace "differences in simulating" with "simulation of"
A: Done.

Q: P18, L17 - reword: "The key difference from the previous study is our use of the ozone"
A: Changed to 'A key difference from the previous study is our use of the use of the ozone'.

Q: P18, L17 - remove "included in our study"
A: Done.

Q: P18, L21 - remove "the" before "non-stomatal"
A: Done.

Q: P18, L22 - should read "To obtain as accurate as possible an estimate "
A: Done.

Q: P18, L23 - replace "it's" with "their"
A: Done.

Q: P18, L24 - replace "considered" with "accounted for"
A: Done.

Q: P18, L25 - suggest moving "(possibly PFT specific)" to come before "flux threshold"
A: Done.

Q: P18, L25 - "it's" should read "its"
A: Changed.

Q: p18, L25 - should the "Y" in "CUOY" be a subscript?
A: No, similar to AOTX the Y is not a subscript.

Q: P18, L32 - insert "see" before "LRTAP"
A: The respective sentence is omitted.

Q: P18, L33 - replace "but only" with "there are" and "exists for" with "of"
A: Done.

Q: P19, L2-4 - What is the implication of this disadvantage to the findings reported here?
A: Two sentences explaining the implications are added: 'This might be an important factor explaining the lower ozone damage estimates of OCN compared to other terrestrial biosphere models. An evaluation of the different proposed damage functions

implemented in terrestrial biosphere models (e.g. Wittig et al. (2007); Lombardozzi et al. (2015); Sitch et al. (2007)) is necessary to elucidate which are able to e.g. reproduce observed patterns of biomass damage and hence might be suitable to predict regional or global damage estimates.'

Q: P19, L5 - replace "damage estimates" with "relationships"
A: Done.

Q: P19, L6 - replace "estimates" with "metrics"
A: Done.

Q: P19, L13 - replace "should be regarded too" with "also requires further analysis"
A: Done.

**3  Conclusion**

Q: This section needs to be substantially expanded. The authors would also do well to identify (even using bullet points if necessary) the key findings of their study and the implications for the land surface and atmosphere research communities. Much of Section 4 could be distilled and included in the Conclusion section.
A: As mentioned above we would like to keep the conclusion short stating briefly the main insights from our work. The Discussion section was shortened to remove redundancy.

Q: P19, L20-1 - replace "to generally consider" with "that"

A: Done.

Q: P19, L21 - reword: "non-stomatal ozone uptake is routinely included in model assessments of ozone damage" and remove "estimate" after "better"
A: The rewording is done. The 'estimate' is not removed since it is an estimate.

Q: P19, L22 - remove "used"
A: Done.

Q: P19, L23 - insert "used here" after "scheme"
A: Done.

Q: P19, L23 - reword: "importance of reliable modelling of canopy conductances as well as realistic"
A: Done.

Q: P19, L24 - insert "as" before "accurate"
A: Done.

Q: P19, L26 - remove "Desirable are"
A: Done.

Q: P19, L27 - insert "are also desirable" after "types"
A: Done.

Q: P19, L29 - replace "regarded" with "considered"

[Figure]

A: Done.

Q: P19, L29 - insert "," after "thresholds"
A: Done.

**4 Appendix**

Q: A P20, L1 - capitalise "Aerodynamic Resistance" and remove "(Appendix material)"
A: Done.

Q: P20, L3 - remove "," after "heights" and replace "This data is" with "These data are"
A: Done.

Q: P20, L4 - replace "in 45m height" with "at 45m"
A: Done.

Q: P20, L7 - what does U10 mean? If at 10m, why is this an appropriate height at which to calculate u*?
A: '$u_{10}$' is now explained as 'from the wind speed at 10 m height ($u_{10}$)'. $u_*$ is assumed to be constant within the surface near atmosphere layer. Since OCN is run offline (not coupled to a climate model) the necessary variables to calculate the friction velocity (e.g. wind speed and aerodynamic resistance) are only available in 10 m height.

Q: P20, L9 - replace "in 45m height" with "at 45m"
A: Done.

Q:Appendix B P20, L21 - Why not use ORCHIDEE to calculate biogenic emissions?
A: OCN was developed from a ORCHIDEE version where biogenic emissions are not calculated. Modules of the current ORCHIDEE can not easily be included in OCN.

Q: P20, L22 - remove "NO from"
A: Done.

Q: P20, L24 - Volcanic emissions of what? Which compounds?
A: Volcanic emissions of $SO_2$ are meant. Respective sentence is changed to: 'Volcanic emissions of sulfur dioxide ($SO_2$) were set to a constant value from the year 2010.'

**5   References**

Please check references carefully.

Q: Tuovinen et al., 2004a and 2004b are the same paper Tuovinen et al., 2009a and 2009b are the same paper
A: This issue is resolved.

**6   Figures**

Q: Throughout - I would suggest that rainbow scale is not the most effective and that limited color graduated scales would be easier to interpret.

A: The color palettes are changes from rainbow to restricted color gradients (palettes from ColorBrewer 2.0).

Q: Fig. 1 Panel (d) - Again, why choose a non-varying measure of LAI (i.e. point samples) rather than MODIS or similar, particularly as you comment on the validity of these measurements for the specific time period modelled? Panel (d) - In its present form this is not a useful panel and I would suggest that it is removed or moved to SI. It distracts from the good fit the model shows to other (more important) variables. Caption - line 4 should read "which are based on point"
A: MODIS data are also subject to a considerable amount of uncertainty. Furthermore the resolution of MODIS data is an additional source of uncertainty. Using observation directly from the site in question seemed to be the most reliable source. We would like to keep panel d) however can remove it or move it to SI when really requested. Caption is changed to "which are based on point".

Q: Fig. 2 x-axis scale - Hours should have a 4-hour or 6-hour scale, not 5. Please state explicitly whether this is local time or UTC. y-axis scale - As the scale is the same across each row I would suggest only one axis scale is required. y-axis scale - for variables that can be negative please add a dashed horizontal line to indicate 0.0; otherwise the axes should cross at zero.
A: X-axis is changed to 3 hour scale (3h - 21h). The time is local time (added to figure caption). Y-axis: the separate scales for each plot secure the readability of the plot. Excluding all but the one in the left column would make it hard to see which values the variable in the other columns take. The minimum for the Y-axes is set to zero.

Q: Fig. 3 scales - please define the scales used in Fig 3 more carefully, either here in the caption or in the appropriate place in the main text. Fig. 4 This figure should be SI. In addition, it is virtually unreadable. I had to view at 600% zoom to make out the yellow and red lines

A: We would like to keep the figure in the main text since it illustrates the robustness of the included deposition module against the exact parameterisation. To make it better readable we skipped the interquartile-range (dark grey area) and stretched the plot. The red and yellow line lie on top of each other. The red line is dashed to show that the yellow line lies directly underneath. Furthermore we added a sentence in the text to explain this fact: 'For all four variables the unperturbed model and the ensemble mean lie on top of each other (see dashed red and yellow line in Fig. 4 a-d).'

Q: Fig. 5 scales - don't use the same colour scales for both absolute values and changes; changes are best shown on blue-red scales. Use e.g. green scale for crop cover.
A: Done.

Q: Fig. 7 scale - please improve the scales; I suggest using a graduated single or limited colour range. panel labels - please use more descriptive panel captions (not just "damage")
A: The color palette is changed.
Regarding the panel label: Since there is only restricted space within the graph corner we choose to state only that damage is plotted and the respective unit which indicates which variable is plotted. In the figure caption it is also stated what is plotted where. To us this seems quite explanatory however we can add also the plotted variable in the corner of the plot what however might overload it.

Q: Fig. 9 To me, this is the KEY figure in this paper. I suggest that you add panels showing changes in CUO from D to D-STO and ATM respectively (giving a 5 panel plot)
A: In our notion the general pattern of a decrease in CUO from ATM to D-STO and D is easy to observe from the present graph. Additional panels showing the exact values seem to add little gain of knowledge. Thus we would like to not include them.

---

## Author Response (AR2)

Dear Editor,

we implemented the changes suggest by the Referee F. Dentener. You can find here the answers to the remarks by F. Dentener and a marked up version of the manuscript showing the implemented changes.

Yours Sincerely
Martina Franz

**1    Answers to Referee F. Dentener**

Q: A remaining issue is the one that was posed in Q1 of my review- I think the discussion on p. 19 is appropriate- perhaps even more clearly the need for a new meta-analysis can be stated that is more specifically targeting the needs of dynamic models.
A: Added on p20 l11: 'Furthermore new damage relationships for different plant groups would be desirable for use in dynamic vegetation models to improve the ozone damage estimates, for example by ensuring an intercept close to one (zero damage at zero accumulated $O_3$).'

**1.1    Some remaining remarks:**

Q: As far as I understand the authors merged figures/tables of a previous appendix into the main text. However, e.g. on page 11 a number of references to appendix figures/tables were retained. Also probably the figures have to be re-ordered in order of appearance.
A: When compiling the *tex file as bgd the Appendix figures are named in arabic numbers as the ones in the main text. When compiling it as bg the Appendix figures are named B1-B3 and it becomes clear which figures belong where. In the text the word Appendix before Fig. was removed for the Appendix figures.

Q: P1l10: 'including ozone damage'. I know I asked this before, but as reader I wonder how sensitive this comparison was to including ozone or not. I guess- given the magnitude of ozone damage- the observational will not really provide strong constraints on ozone damage- but we can argue that having these OK is of course a good 'baseline' on top of which ozone damage

can be evaluated.

A: Added: 'This comparison provides a good baseline on top of which ozone damage can be evaluated.'

Q: P2l10 the major causes are
A: Changed.

Q: P2 l11 also mention VOCs (volatile organic carbons) locally more important than CO.
A: VOCs are included into the respective sentence.

Q: P2 l30 AOT is not so much used in the US. I propose to say: In Europe, the standard method . . . .
A: Done.

Q: P3 l. 2 .. of one species? What is meant?
A: It is meant that canopy conductance differs between species but also between regional provenances of species. The 'one' was deleted.

Q: P4 l.21: OCN stands for Orchidee Carbon Nitrogen?
A: OCN is a further development of ORCHIDEE. In the respective sentence a 'O' and 'C' is added after ORCHIDEE and carbon to make this more clearly.

Q: P8 l.12: 30-50 add units.
A: Done.

[revised manuscript text omitted]